# ONLINE STABILIZATION OF SPIKING NEURAL NETWORKS

**Yaoyu Zhu**[1], **Jianhao Ding**[1], **Tiejun Huang**[1,2], **Xiaodong Xie**[1] **& Zhaofei Yu**[1,2] *
[1] School of Computer Science, Peking University
[2] Institute for Artificial Intelligence, Peking University

## ABSTRACT

Spiking neural networks (SNNs), attributed to the binary, event-driven nature of spikes, possess heightened biological plausibility and enhanced energy efficiency on neuromorphic hardware compared to analog neural networks (ANNs). Mainstream SNN training schemes apply backpropagation-through-time (BPTT) with surrogate gradients to replace the non-differentiable spike emitting process during backpropagation. While achieving competitive performance, the requirement for storing intermediate information at all time-steps incurs higher memory consumption and fails to fulfill the online property crucial to biological brains. Our work focuses on online training techniques, aiming for memory efficiency while preserving biological plausibility. The limitation of not having access to future information in early time steps in online training has constrained previous efforts to incorporate advantageous modules such as batch normalization. To address this problem, we propose Online Spiking Renormalization (OSR) to ensure consistent parameters between testing and training, and Online Threshold Stabilizer (OTS) to stabilize neuron firing rates across time steps. Furthermore, we design a novel online approach to compute the sample mean and variance over time for OSR. Experiments conducted on various datasets demonstrate the proposed method's superior performance among SNN online training algorithms. Our code is available at https://github.com/zhuyaoyu/SNN-online-normalization.

## 1 INTRODUCTION

Regarded as the third generation of neural networks, spiking neural networks (SNNs) possess a greater level of biological plausibility (Zenke et al., 2021) than their second generation counterparts – analog neural networks (ANNs) due to the binary and event-driven nature of spikes. The binary nature of spikes in SNNs eliminates the need for multiplication during inference, leading to improved energy efficiency when deployed on neuromorphic hardware (Furber et al., 2014; Merolla et al., 2014; Shen et al., 2016; Davies et al., 2018; Pei et al., 2019). However, the discontinuity of binary spikes also poses challenges in the training of SNNs.

To address the non-differentiable issue associated with the spike emitting process in SNN training, various approaches have been proposed. The mainstream direct training techniques use surrogate gradients to address this problem, which replaces the non-differentiable Heaviside function during the spike firing process with a differentiable surrogate function (Neftci et al., 2019). In addition to this, they just regard SNNs as binary recurrent neural networks (RNNs) and use backpropagation-through-time (BPTT) for SNN training (Bellec et al., 2018; Zenke & Ganguli, 2018; Wu et al., 2018). Although competitive performances are achieved on the CIFAR-10/100 and ImageNet datasets (Deng et al., 2021; Fang et al., 2021) with a relatively short simulation time, these methods require storing intermediate information of all time-steps for gradient backpropagation. An alternative approach to train SNNs is to use the assistance of ANNs. Several works first train ANNs and

---

*Corresponding author: yuzf12@pku.edu.cn

then convert them to SNNs (Cao et al., 2015; Rueckauer et al., 2017; Han et al., 2020; Bu et al., 2022a; Deng & Gu, 2021; Bu et al., 2022b). However, these methods often require a longer simulation time and result in more fired spikes. The long simulation time will lead to high latency, while more fired spikes will consume more energy. Overall, these approaches bring about extra expenses either in the training phases or in the testing phases while not satisfying the online property of the learning process in biological brains.

Recently, online training techniques have been developed to save memory costs while maintaining the biologically plausible online property during the training process. However, the limitation of not having access to future information in the early time steps has constrained previous efforts to incorporate advantageous modules such as batch normalization (BN). In this work, we design a mechanism that bypasses the need for future information while maintaining consistency across time-steps, thereby reducing the overfitting problem associated with treating different time-steps with different BN. Our main contributions can be summarized as follows:

1. We propose Online Spiking Renormalization (OSR), ensuring consistent scale and shift parameters between testing and training. This helps eliminate the normalization parameter difference when applying BN separately for each time-step. In addition, we introduce an online approach for computing a variable's all-time mean and variance that dynamically changes over time for OSR.
2. We devise Online Threshold Stabilizer (OTS), aiming at stabilizing neuron firing rates across varying time steps, which also effectively regulates the overall firing rate.
3. We conduct experiments on CIFAR10, CIFAR100, CIFAR10-DVS, DVS-Gesture, and Imagenet datasets and demonstrate that our proposed method achieves state-of-the-art performance among SNN online training algorithms.

## 2 RELATED WORK

### 2.1 ONLINE TRAINING APPROACHES

Online training allows real-time parameter updates as new data arrives, especially useful for RNNs and SNNs spanning multiple time-steps. This mechanism serves to curtail memory usage, a particularly advantageous feature when dealing with many time-steps.

Existing literature on RNNs has delved into various approaches to online learning. Real-time recurrent learning (RTRL), introduced by Williams & Zipser (1989), propagates partial derivatives of hidden states across parameters throughout time, enabling the computation of gradients in a forward-in-time manner. Many recent research endeavors, exemplified by UORO (Tallec & Ollivier, 2017), KF-RTRL (Mujika et al., 2018), and SnAp (Menick et al., 2020), have explored enhancing the memory and time efficiency of RTRL through tailored approximations for more pragmatic utilization. Another work put forward a proposition to update parameters in an online fashion, utilizing decoupled gradients coupled with regularization at each time-step (Kag & Saligrama, 2021).

In the domain of SNNs, numerous studies have drawn inspiration from online training techniques developed for RNNs. Some of these works adopt the fundamental principles of RTRL and tailor them to streamline the training process for SNNs (Zenke & Ganguli, 2018; Bellec et al., 2020; Bohnstingl et al., 2022). Yin et al. (2022) directly applied the approach proposed by Kag & Saligrama (2021) to train SNNs. Zenke & Ganguli (2018) connected the online learning rule for leaky integrate-and-fire (LIF) neurons with the nonlinear Hebbian three-factor rule, and Kaiser et al. (2020) extended the neuron model to a double-exponential spike-response model. Xiao et al. (2022) successfully extended online training methodologies to accommodate large-scale tasks such as the ImageNet classification. However, all these works did not consider incorporating network modules like batch normalization to enhance the network performance. As a result, they suffer from a performance disadvantage compared to their BPTT counterparts.

## 2.2 NORMALIZATION MECHANISMS

Normalization mechanisms are commonly used in neural networks to stabilize network training, which speeds up convergence and enhances network performance. Typical normalization techniques include batch normalization (BN) (Ioffe & Szegedy, 2015), instance normalization (IN) (Ulyanov et al., 2016), group normalization (GN) (Wu & He, 2018), and layer normalization (LN) (Ba et al., 2016). A subsequent work, batch renormalization (Ioffe, 2017), improves BN by eliminating the difference between the batch mean and variance between the training and testing phases.

In spiking neural networks, researchers have also tried to incorporate normalization techniques to enhance SNN performance. For instance, Kim & Panda (2021) proposed BNTT to regulate firing rates by utilizing separate BN parameters at different time steps. Zheng et al. (2021) proposed threshold-dependent batch normalization (tdBN), which extends the scope of BN to the additional temporal dimension and takes into account the impact of threshold on firing rates. TEBN (Duan et al., 2022) combined elements from both of these approaches by applying BN across the spatial-temporal dimension, while utilizing separate scale and shift parameters at different time steps. Although most works apply normalization to the input current, some studies explore normalization for other variables. For example, PSP-BN (Ikegawa et al., 2022) used unique statistics, the second raw moment of post-synaptic potential, as the denominator for normalization, which can be inserted right after the spiking functions. This approach leads to a higher complexity of BN parameters and the potential risk of breaking the temporal coherence of information. Among the aforementioned works, the most successful ones (Duan et al., 2022; Zheng et al., 2021) used information from all time-steps for BN. However, these methods cannot be directly applied to online learning.

## 3 PRELIMINARIES

### 3.1 LEAKY INTEGRATE AND FIRE NEURON

Spiking neurons are the basic building blocks of SNNs, with the LIF neuron model being the most commonly used (Gerstner et al., 2014). The dynamic of the LIF neuron before firing can be described by:

$$\tau \frac{du(t)}{dt} = -(u(t) - u_{\text{rest}}) + I(t), \tag{1}$$

where $u(t)$ is the membrane potential of the neuron at time $t$, $I(t)$ is the input current received by the neuron, $\tau$ is the membrane time constant, and $u_{\text{rest}}$ is the resting potential. When membrane potential $u(t)$ reaches a certain threshold $\theta$, the neuron will emit a spike, and $u(t)$ will be suddenly reset to a value $u_{\text{reset}}$. In practice, we often use a discrete form of Eq. 1, which can be represented as:

$$\boldsymbol{u}^l[t] = (1 - \frac{1}{\tau^l})\boldsymbol{u}^l[t - 0.5] + \boldsymbol{W}^l \boldsymbol{s}^{l-1}[t], \tag{2}$$

$$\boldsymbol{s}^l[t] = \Theta(\boldsymbol{u}^l[t] - \theta), \tag{3}$$

$$\boldsymbol{u}^l[t + 0.5] = \boldsymbol{u}^l[t] \odot (1 - \boldsymbol{s}^l[t]). \tag{4}$$

Here, we use a tensor form that $\boldsymbol{u}^l$, $\boldsymbol{W}^l$, and $\boldsymbol{s}^l$ denote the membrane potential tensor, weight matrix between layers $l - 1$ and $l$, output spike tensor of layer $l$, respectively. Among them, $\boldsymbol{u}^l[t]$ is the membrane potential after decay and adding input but before the reset, and $\boldsymbol{u}^l[t + \frac{1}{2}]$ is the membrane potential after reset. $\Theta$ is a Heaviside step function. The element of $\boldsymbol{s}^l[t]$ equals 1 if the neuron fires and 0 otherwise.

## 4 METHODS

Our holistic method is illustrated in Figure 1. In the following parts, we first briefly introduce the forward and backward propagation processes of our algorithm and then elaborate on the modules we add to the network.

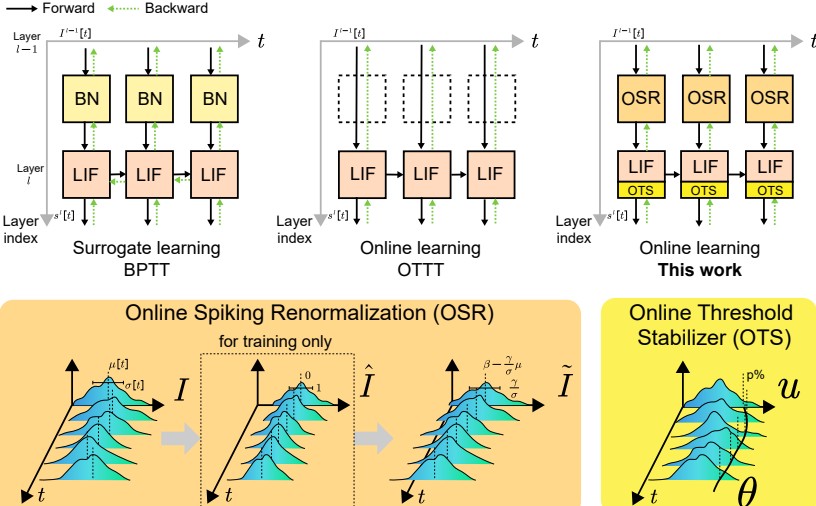

Figure 1: Illustration of online stabilization techniques for SNN. Our method uses online spiking renormalization to improve the generalization and an online threshold stabilizer to regulate the firing rate within the framework of online SNN training, which requires less memory usage than BPTT training. OTTT (Xiao et al., 2022) adopts normalization-free networks and thus has no BN modules.

The major modules are online spiking renormalization (OSR) and online threshold stabilizer (OTS). Besides, an online calculation method of all-time mean and variance is introduced in OSR.

## 4.1 FORWARD AND BACKWARD PROPAGATION

In the forward stage, our method uses the LIF formulas (Eqs. 2-4). The OSR replaces $s^{l-1}[t]\boldsymbol{W}^l$ with $renorm(s^{l-1}[t]\boldsymbol{W}^l)$ in Eq. 2, and the OTS changes threshold $\theta$ in Eq. 3 over time-steps.

In the backward stage, we select the TET loss (Deng et al., 2021) as our loss function since the loss function needs to provide feedback at each time-step:

$$\mathcal{L} = \frac{1}{T}\sum_{t=1}^{T}\mathcal{L}_t = \frac{1}{T}\sum_{t=1}^{T}\left((1-\epsilon)\sum_{i=1}^{n}y_i \log o_i[t] + \epsilon\sum_{i=1}^{n}(o_i[t] - \phi(y_i))^2\right), \tag{5}$$

where $T$ is the total simulation time, $y_i$ denotes whether the label is equal to $i$, and $o_i[t]$ is the spikes of output neuron $i$ at time $t$ in the output layer. An additional MSE loss is introduced as a regularization term (as proposed by Deng et al. (2021)) with weight $\epsilon$, and $\phi(y_i)$ is the target value of $y_i$ set in MSE loss.

For the online gradient propagation, we remove the propagation path of neuron membrane potential decay and reset (from the next time step to the last time step) for membrane potential (as shown in Figure 1). Then the gradients received by membrane potential and weights become:

$$\frac{\partial \mathcal{L}}{\partial u_y^l[t]} = \frac{\partial \mathcal{L}_t}{\partial s_y^l[t]}\frac{\partial s_y^l[t]}{\partial u_y^l[t]}, \tag{6}$$

$$\frac{\partial \mathcal{L}}{\partial w_{xy}^l} = \sum_{t=1}^{T}\frac{\partial \mathcal{L}_t}{\partial s_y^l[t]}\frac{\partial s_y^l[t]}{\partial u_y^l[t]}\frac{\partial u_y^l[t]}{\partial w_{xy}^l}. \tag{7}$$

Note that Eq. 7 just sums up the derivative of $\mathcal{L}_t$ to $w_{xy}^l$ at all time-steps, so there is no backward or forward temporal dependency both in Eqs. 6 and 7)

## 4.2 INCORPORATING BATCH NORMALIZATION INTO ONLINE ALGORITHMS

In Zheng et al. (2021); Kim & Panda (2021); Duan et al. (2022), it is shown that applying BN once across all time-steps, rather than separately on each time-step, yields superior performance. However, in online training, it is impractical as we need normalization before having information on all time-steps. As per Duan et al. (2022), using mean and variance across all time-steps is crucial for reducing temporal covariate shift and enhancing performance. A key feature of normalizing by the global mean and variance is that, the transformation of all time-steps are the same during normalization. Therefore, a question naturally arises: can we normalize inputs at all time-steps with the same mean and variance when we do not have the all-time data?

**Online Spiking Renormalization (OSR).** Although we do not have the whole data of the current batch, we have data from previous batches and can apply BN transformation at all time-steps based on these data. The running mean $\hat{\mu}$ and running variance $\hat{\sigma^2}$ is a good choice. Using them as the normalization parameter brings an additional benefit: The BN transformation will be the same between the training stage and the inference stage. Specifically, we apply the transformation

$$\tilde{\boldsymbol{I}}[t] = \gamma \cdot \frac{\boldsymbol{I}[t] - \hat{\mu}}{\sqrt{\hat{\sigma^2} + \epsilon}} + \beta \tag{8}$$

during the forward stage in training, where $\boldsymbol{I}[t]$ is the neurons' input currents to be normalized. The next question is: How to compute gradients in the backward stage if we use this forward transformation? The $\hat{\mu}$ and $\hat{\sigma^2}$ come from previous data instead of the current batch data. Therefore, if no additional mechanisms are involved, this 'standardization' just plays the role of linear transformation instead of real normalization. Our solution is online spiking renormalization (OSR), which first applies a real normalization and then unifies transformation among time-steps by another linear transform. To be specific, we first normalize $\boldsymbol{I}[t]$ to $\hat{\boldsymbol{I}}[t] = \frac{\boldsymbol{I}[t] - \mu[t]}{\sqrt{\sigma^2[t] + \epsilon}}$ and then linearly transform it twice to $\tilde{\boldsymbol{I}}[t] = \gamma \cdot \frac{\boldsymbol{I}[t] - \hat{\mu}}{\sqrt{\hat{\sigma^2} + \epsilon}} + \beta$:

$$\tilde{\boldsymbol{I}}[t] = \gamma \cdot \frac{\boldsymbol{I}[t] - \hat{\mu}}{\sqrt{\hat{\sigma^2} + \epsilon}} + \beta = \gamma \cdot \left( \hat{\boldsymbol{I}}[t] \cdot \frac{\sqrt{\sigma^2[t] + \epsilon}}{\sqrt{\hat{\sigma^2} + \epsilon}} + \frac{\mu[t] - \hat{\mu}}{\sqrt{\hat{\sigma^2} + \epsilon}} \right) + \beta. \tag{9}$$

Eq. 9 denotes the normalization followed by a linear transformation. The gradients for $\boldsymbol{I}[t], \gamma, \beta$ are:

$$\frac{\partial \mathcal{L}}{\partial \boldsymbol{I}[t]} = \frac{\partial \mathcal{L}}{\partial \tilde{\boldsymbol{I}}[t]} \cdot \frac{\partial \hat{\boldsymbol{I}}[t]}{\partial \boldsymbol{I}[t]} \cdot \gamma \cdot \frac{\sqrt{\sigma^2[t] + \epsilon}}{\sqrt{\hat{\sigma^2} + \epsilon}}, \tag{10}$$

$$\frac{\partial \mathcal{L}}{\partial \gamma} = \sum_x \frac{\partial \mathcal{L}}{\partial \tilde{I}_x[t]} \left( \hat{I}_x[t] \cdot \frac{\sqrt{\sigma^2[t] + \epsilon}}{\sqrt{\hat{\sigma^2} + \epsilon}} + \frac{\mu[t] - \hat{\mu}}{\sqrt{\hat{\sigma^2} + \epsilon}} \right), \tag{11}$$

$$\frac{\partial \mathcal{L}}{\partial \beta} = \sum_x \frac{\partial \mathcal{L}}{\partial \tilde{I}_x[t]}. \tag{12}$$

**Online Calculation of All-time Mean and Variance.** In OSR, the running mean $\hat{\mu}$ and running variance $\hat{\sigma^2}$ are the running average of all-time mean $\mu$ and variance $\sigma^2$ of a batch. To keep the memory cost low, we need to calculate these all-time statistics in an online fashion, utilizing the mean and variance of each time-step: $\mu[1], \cdots, \mu[T]$ and $\sigma^2[1], \cdots, \sigma^2[T]$. Their relationship can be described by the following equations:

$$\mu[t] = \frac{1}{m} \sum_{x=1}^{m} I_x[t], \qquad \sigma^2[t] = \frac{1}{m} \sum_{x=1}^{m} (I_x[t] - \mu[t])^2, \tag{13}$$

$$\mu = \frac{1}{mT} \sum_{t=1}^{T} \sum_{x=1}^{m} I_x[t] = \frac{1}{T} \sum_{t=1}^{T} \mu[t], \tag{14}$$

$$\sigma^2 = \frac{1}{mT} \sum_{t=1}^{T} \sum_{x=1}^{m} (I_x[t] - \mu)^2 = \frac{1}{T} \sum_{t=1}^{T} \sigma^2[t] + \frac{1}{T} \sum_{t=1}^{T} \mu[t]^2 - \mu^2. \tag{15}$$

Hence, we can initialize $\mu$ and $\sigma^2$ as 0 for each batch, add $\frac{1}{T}\mu[t]$ to $\mu$ and add $\frac{1}{T}(\sigma^2[t] + \mu[t]^2)$ to $\sigma^2$ at each time step, and subtract $\mu^2$ from $\sigma^2$ at the last time step.

**Online Threshold Stabilizer (OTS).** To enhance the stability of mean and variance in the OSR process during training, we introduce the OTS mechanism. The variable subject to normalization is the input current of neurons, and our objective is to ensure the mean and variance of it remain stable across all time-steps. This raises a question: When should we intervene to stabilize the mean and variance of input currents?

The mean and variance of the input current in a layer are significantly influenced by the output spikes from the preceding layer, making it essential to stabilize the firing rate of each layer. The firing rate is determined by the proportion of membrane potential surpassing the firing threshold within discrete time-steps. Consequently, we can adjust either the membrane potential or the firing threshold to regulate the firing rate. Between these options, regulating the firing threshold stands out as a judicious choice: it leaves the neuronal dynamics unchanged and only impacts backward propagation by altering the values of the surrogate function.

Specifically, we assume the membrane potential of neurons in one layer at time $t$ follows a normal distribution $N(\mu_{\text{mem}}[t], \sigma^2_{\text{mem}}[t])$ (where we denote $\theta[t]$, $\mu_{\text{mem}}[t]$, and $\sigma_{\text{mem}}[t]$ are the threshold, mean of membrane potential, and variance of membrane potential at time $t$), then the firing rate of this layer at time $t$ is

$$1 - \Phi^{-1} \left( \frac{\theta[t] - \mu_{\text{mem}}[t]}{\sigma_{\text{mem}}[t]} \right), \tag{16}$$

where $\Phi(x) = \frac{1}{\sqrt{2\pi}} \int_{-\infty}^{x} e^{-\frac{y^2}{2}} dy$ is the cumulative distribution function of normal distribution. To keep this ratio constant among time-steps, we need to keep the quantile $\frac{\theta[t] - \mu_{\text{mem}}[t]}{\sigma_{\text{mem}}[t]}$ constant. Under this control, the adjusted threshold at time $t$, $\theta[t] = \mu_{\text{mem}}[t] + \sigma_{\text{mem}}[t] \cdot \frac{\theta[1] - \mu_{\text{mem}}[1]}{\sigma_{\text{mem}}[1]}$.

The overall algorithm description is provided in Appendix B.

### 4.3 THEORETICAL ANALYSIS

In this section, we discuss how our online threshold stabilizer (OTS) helps stabilize online spiking renormalization (OSR). The process involves three stages: adjusting the firing threshold of layer $l - 1$, corresponding adjustment of the firing rate of layer $l - 1$, and regulating the mean and variance before normalization in layer $l$ to ensure stability. This involves two crucial aspects: adjusting the threshold for a stable firing rate, which subsequently stabilizes the mean and variance. For the step from threshold to firing rate, existing research has shown that when a LIF neuron receives constant input with Gaussian noise, the membrane potential will have a Gaussian distribution (Hohn & Burkitt, 2001). This implies the reasonableness of our Gaussian distribution assumption of membrane potential in OTS, further supporting its process. For the step from firing rate to mean and variance, studying the property of the all-time sample variance $\sigma^2$ is a good choice: In Eq. 15, $\sigma^2$ can be split into two parts: The first part is $\frac{1}{T} \sum_{t=1}^{T} \sigma^2[t]$, which stands for the average variance inside each time-step. The second part is $\frac{1}{T} \sum_{t=1}^{T} \mu[t]^2 - \mu^2$, which is the variance of the mean at each time-step (variance of $\mu[1], \cdots, \mu[T]$). To stabilize the whole training process, we want the mean among different time-steps to vary as little as possible. In other words, we want the variance of the mean among time-steps to be low. On the other hand, for variance inside each time-step, we do not need it to be low.

Denote $p[t]$ to be the firing probability of each neuron at time-step $t$ and gross firing probability $p = \frac{1}{T} \sum_{i=1}^{T} p[t]$. To proceed with the theoretical derivation, we must establish the following assumptions:

**Assumption 4.1.** Assume all entries of $s^{l-1}[t]$ (of size $B \cdot C_{in}$) and $W^l$ (of size $C_{in} \cdot C_{out}$) are independent for $1 \leq t \leq T$, all $s_i^{l-1}[t]$ obey i.i.d Bernoulli($p[t]$) distribution, and all $w_{ji}^l$ obey any i.i.d distribution.

Under the above assumptions, we have the following conclusions (note we only discuss the expectation of the target variables since both the sample mean $\mu$ and the sample variance $\sigma^2$ are estimated statistics):

**Theorem 4.2.** *When Assumption 4.1 holds and the gross firing rate $p$ holds constant, then the expectation of sample variance of $\mu[t]$ among time-steps $\mathbb{E}\left[\frac{1}{T}\sum_{t=1}^{T}\mu[t]^2 - \mu^2\right]$ increases when the variance of firing rate among time-steps $\frac{1}{T}\sum_{t=1}^{T}p[t]^2 - p^2$ increases.*

**Theorem 4.3.** *When Assumption 4.1 holds and the gross firing rate $p$ keeps constant, then the expectation of variance within time-steps $\mathbb{E}\left[\frac{1}{T}\sum_{t=1}^{T}\sigma^2[t]\right]$ keeps constant.*

The detailed proof is provided in Appendix A. These results indicate that given the gross firing rate ($p$) constant, reducing the variance of firing probability ($p[t]$) among time-steps will reduce the variance of the mean ($\mu[t]$) among time-steps (Theorem. 4.2) but will not affect the variance inside time-steps ($\sum \sigma^2[t]$) (Theorem. 4.3). Thus, a steady firing rate helps stabilize the sample mean, which further indicates that our OTS mechanism helps our OSR mechanism. Related experimental results are shown in the ablation study.

## 5 EXPERIMENTS

To show the effectiveness of our proposed method, we conduct experiments on CIFAR10, CIFAR100 (Krizhevsky et al., 2009), DVS-Gesture (Amir et al., 2017), CIFAR10-DVS (Li et al., 2017), and Imagenet (Deng et al., 2009) datasets to evaluate the performance of our method. The model we choose is consistent with OTTT (Xiao et al., 2022) to conduct a fair comparison. All experiments are run on Nvidia RTX 4090 GPUs with Pytorch 2.0. The implementation details are provided in Appendix C.

### 5.1 COMPARISON WITH OTHER WORKS

Here we compare our approach with previous SNN training methods. We select the BPTT-based algorithms tdBN (Zheng et al., 2021), SEW (Fang et al., 2021), TET (Deng et al., 2021), TEBN (Duan et al., 2022), and an online algorithm OTTT (Xiao et al., 2022). The results have shown that our algorithm performs well on all datasets. For the CIFAR10 dataset, we have outperformed tdBN, OTTT, and TET. For the CIFAR100 dataset, we have outperformed TET and OTTT. For the DVS-Gesture dataset, we have outperformed all listed methods, including tdBN and OTTT. For the CIFAR10-DVS dataset, we have outperformed tdBN and OTTT. For the Imagenet dataset, we have outperformed tdBN and OTTT. Note that the network that OTTT uses (NF-Resnet-34) adds membrane potential in the shortcut connection, which enhances its overall performance over SEW-Resnet-34 and Resnet-34. We test our method for the same architecture (the last line) and achieve better performance with fewer time-steps. Among the online algorithms, we have outperformed OTTT on all datasets with fewer time-steps ($T = 4$ vs $T = 6$). In addition, although the overall performance of state-of-the-art BPTT-based algorithms outperforms the online ones in Table 1, they require more memory, especially when the number of total time steps is large (the detailed information is provided in Section 5.2).

**Comparison with Vanilla BN.** To show the necessity of our approach, we compare it with a vanilla BN, which applies BN each time-step solely based on data from that time-step. The result is shown in the BN (vanilla) line for the Imagenet dataset, and we can see our approach outperforms this vanilla BN by around 4%. More detailed ablation studies for OSR and OTS on various datasets are provided in Appendix D.

Table 1: Performance comparison on CIFAR-10/100, DVS-Gesture, CIFAR10-DVS, and Imagenet

| Dataset | Model | Online or not | Architecture | Time steps | Accuracy |
|---|---|---|---|---|---|
| CIFAR10 | tdBN (Zheng et al., 2021) | ✗ | Resnet-19 | 4 | 92.92% |
| | TET (Deng et al., 2021) | ✗ | Resnet-19 | 4 | 94.44% |
| | TEBN (Duan et al., 2022) | ✗ | Resnet-19 | 4 | 95.58% |
| | OTTT (Xiao et al., 2022) | ✔ | VGGSNN | 6 | 93.58% |
| | **Ours** | ✔ | VGGSNN | 4 | 94.35% |
| | | ✔ | Resnet-19 | 4 | 95.20% |
| CIFAR100 | TET (Deng et al., 2021) | ✗ | Resnet-19 | 4 | 74.47% |
| | TEBN (Duan et al., 2022) | ✗ | Resnet-19 | 4 | 78.71% |
| | OTTT (Xiao et al., 2022) | ✔ | VGGSNN | 6 | 71.11% |
| | **Ours** | ✔ | VGGSNN | 4 | 76.48% |
| | | ✔ | Resnet-19 | 4 | 77.86% |
| DVS-Gesture | tdBN (Zheng et al., 2021) | ✗ | Resnet-17 | 40 | 96.88% |
| | OTTT (Xiao et al., 2022) | ✔ | VGGSNN | 20 | 96.88% |
| | **Ours** | ✔ | VGGSNN | 20 | 97.57% |
| CIFAR10-DVS | tdBN (Zheng et al., 2021) | ✗ | Resnet-19 | 10 | 67.80% |
| | TET (Deng et al., 2021) | ✗ | VGG-11 | 10 | 83.17% |
| | TEBN (Duan et al., 2022) | ✗ | VGGSNN | 10 | 84.90% |
| | OTTT (Xiao et al., 2022) | ✔ | VGGSNN | 10 | 76.30% |
| | **Ours** | ✔ | VGGSNN | 10 | 82.40% |
| Imagenet | tdBN (Zheng et al., 2021) | ✗ | Resnet-34 | 6 | 63.72% |
| | SEW (Fang et al., 2021) | ✗ | SEW-Resnet-34 | 4 | 67.04% |
| | TET (Deng et al., 2021) | ✗ | SEW-Resnet-34 | 4 | 68.00% |
| | TEBN (Duan et al., 2022) | ✗ | SEW-Resnet-34 | 4 | 68.28% |
| | OTTT (Xiao et al., 2022) | ✔ | NF-Resnet-34 | 6 | 65.15% |
| | BN(Vanilla) | ✔ | SEW-Resnet-34 | 4 | 60.48% |
| | BN(OSR+OTS)(**Ours**) | ✔ | SEW-Resnet-34 | 4 | 64.14% |
| | BN(OSR+OTS)(**Ours**) | ✔ | NF-Resnet-34* | 4 | 67.54% |

* To keep consistent with OTTT, we use the name 'NF-Resnet' here to represent adding membrane potential in the shortcut connection in Resnet. Note that 'NF' in OTTT stands for normalizer-free, but the corresponding part (weight standardization and scaling factors $\alpha, \beta$ along with the corresponding operations to keep variance stable) of this network is eliminated in our work.

## 5.2 QUALITATIVE RESULTS

**A. Memory Usage:** We compare the training memory usage between online algorithms and BPTT algorithms here. We test the case where $T = 2, 4, 6, 8, 10, 15, 20, 25, 30$ on the CIFAR10 dataset with VGGSNN architecture and a batch size of 128. The memory usage statistics are plotted in Figure 2 (a). We can see that our method maintains a constant memory requirement irrespective of time-steps, whereas BPTT approaches scale memory usage linearly with the number of time-steps. In addition, even when the number of time-steps is as low as 2, the memory cost of our algorithm is still lower than that of its BPTT counterpart.

**B. Firing Rate Statistics:** We compare the firing rate statistics among different configurations of our proposed modules. We test these statistics on Imagenet, using the SEW-Resnet-34 architecture with total time-steps $T = 4$. The gross firing rate statistics are listed in Table 2 and the per-time-step firing rates are plotted in Figure 2 (b). Results have shown that OTS successfully decreases the gross firing rate, which meets our expectations since it raises the thresholds in the latter time-steps. The effect of OSR on firing rates is more

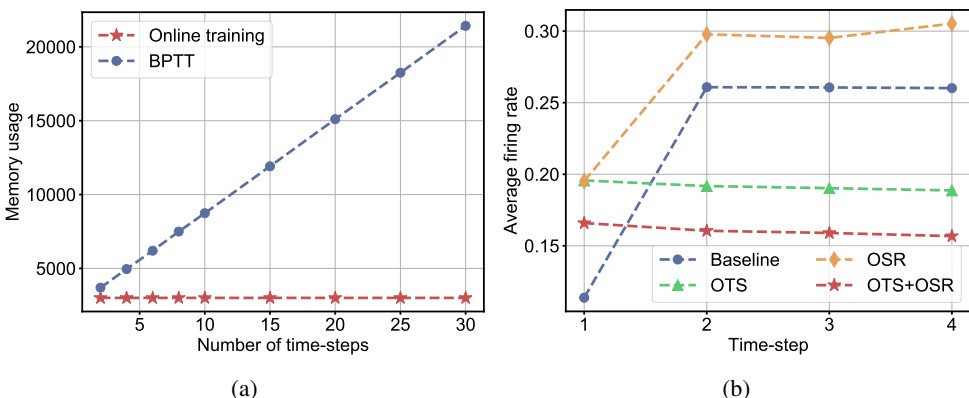

(a)  (b)

Figure 2: (a) Comparison of memory usage between our method and BPTT. BPTT incurs memory costs linearly proportional to time-steps, whereas our approach maintains constant memory usage regardless of time-steps. (b) Firing rate statistics of different configurations. From the figures, we know that the online threshold stabilizer indeed stabilizes the firing rate among time-steps.

Table 2: Gross firing rate

| Configuration | OTTT (Xiao et al., 2022) | Vanilla BN | OSR | OTS | OSR+OTS |
|---|---|---|---|---|---|
| Gross firing rate | 24% | 22.39% | 27.34% | 19.16% | 16.06% |

interesting: When OTS is not added, it increases the gross firing rate. However, it decreases the total firing rate when OTS is added. For per-time-step firing rates, when the OTS mechanism is not added, the neurons fire far fewer spikes in the first time-step compared with later time-steps, while the firing rate is relatively stable from the second time-step to the last time-step. Besides, our OTS mechanism has greatly alleviated but not perfectly eliminated the firing rate variation among time-steps. It slightly over-lifts the firing rate of the first time-step, which might be caused by the firing rate distribution difference among time-steps.

## 6 CONCLUSION AND FUTURE WORK

In this paper, we investigate online training for spiking neural networks, aiming to reduce training memory costs. We integrate essential batch normalization into the online training process by introducing online spiking renormalization and online threshold stabilizers to enhance training stability. Experiments on diverse datasets demonstrate the effectiveness of our proposed modules, showcasing the superior performance of our holistic approach among SNN online training algorithms. However, our approach currently falls short of BPTT in performance, primarily due to the absence of inner-layer and inter-layer reverse-in-time dependencies during backpropagation. Addressing the inner-layer dependency might involve incorporating eligibility traces, but effectively managing the significant inter-layer dependency in online learning remains a challenge. Moreover, achieving biologically plausible learning necessitates local (Journé et al., 2022) and event-driven (Zhu et al., 2022) properties in addition to online behavior—areas we haven't extensively explored in this work. These shortcomings present promising avenues for future research and deeper investigation.

## ACKNOWLEDGEMENTS

This work was supported by the National Natural Science Foundation of China(62176003, 62088102) and by Beijing Nova Program (20230484362).

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

## A  THEORETICAL DERIVATION

To make it easy to follow, we write Assumption. 4.1 again in the following:

**Assumption A.1.** Assume all entries of $s^{l-1}[t]$ (of size $B \cdot C_{in}$) and $W^l$ (of size $C_{in} \cdot C_{out}$) are independent for $1 \leq t \leq T$, all $s_i^{l-1}[t]$ obey i.i.d Bernoulli($p[t]$) distribution, and all $w_{ji}^l$ obey any i.i.d distribution.

For simplicity, we omit the superscript $l$ and $l-1$ in the following derivation. Before proving the theorems, we derive the mean and variance of variable $I_{bi}[t]$ in Lemma. A.2 and Lemma. A.3:

**Lemma A.2.** *When Assumption 4.1 holds, all $I_{bi}[t]$ will share the identical distribution, and $\mathbb{E}[I_{bi}[t]] = C_{in}p[t]\mathbb{E}[w_{ji}]$, $\mathbb{VAR}[I_{bi}[t]] = C_{in}(p[t]\mathbb{VAR}[w_{ji}] + (p[t] - p[t]^2)\mathbb{E}^2[w_{ji}])$.*

*Proof.* Since $I_{bi}[t] = \sum_{j=1}^{C_{in}} s_{bj}[t]w_{ji}$ are all sum of products of independent variables with identical distributions ($s_{bj}[t]$ and $w_{ji}$), they share the identical distribution. We calculate the mean and variance of $I_{bi}[t]$ as follows:

$$\mathbb{E}[I_{bi}[t]] = \sum_{j=1}^{C_{in}} \mathbb{E}(s_{bj}[t])\mathbb{E}(w_{ji}) = C_{in}p[t]\mathbb{E}[w_{ji}] \tag{17}$$

$$\mathbb{E}[I_{bi}^2[t]] = \mathbb{E}\left[\left(\sum_{j=1}^{C_{in}} s_{bj}[t]w_{ji}\right)^2\right] = \mathbb{E}\left[\sum_{j=1}^{C_{in}} \left(s_{bj}[t]w_{ji}\right)^2\right] + C_{in}(C_{in}-1)p[t]^2\mathbb{E}^2[w_{ji}]$$

$$= \sum_{j=1}^{C_{in}} \mathbb{E}[s_{bj}[t]^2]\mathbb{E}[w_{ji}^2] + C_{in}(C_{in}-1)p[t]^2\mathbb{E}^2[w_{ji}]$$

$$= C_{in}p[t]\mathbb{E}[w_{ji}^2] + C_{in}(C_{in}-1)p[t]^2\mathbb{E}^2[w_{ji}] \tag{18}$$

$$\mathbb{VAR}(I_{bi}[t]) = \mathbb{E}[I_{bi}^2[t]] - \mathbb{E}^2[I_{bi}[t]] = C_{in}(p[t]\mathbb{E}[w_{ji}^2] - p[t]^2\mathbb{E}^2[w_{ji}])$$

$$= C_{in}(p[t]\mathbb{VAR}[w_{ji}] + (p[t] - p[t]^2)\mathbb{E}^2[w_{ji}]) \tag{19}$$

$\square$

**Lemma A.3.** *When Assumption 4.1 holds, for all $1 \leq b_1, b_2 \leq B$, $1 \leq i_1, i_2 \leq C_{out}$, and $1 \leq t_1, t_2 \leq T$, $I_{b_1 i_1}[t_1]$ and $I_{b_2 i_2}[t_2]$ are uncorrelated when $(b_1, t_1) \neq (b_2, t_2)$ and $i_1 \neq i_2$. When $(b_1, t_1) = (b_2, t_2)$ and $i_1 \neq i_2$, $\mathbb{COV}(I_{bi_1}[t], I_{bi_2}[t]) = C_{in}\mathbb{E}^2[w_{ji}](p[t] - p[t]^2)$; When $(b_1, t_1) \neq (b_2, t_2)$ and $i_1 = i_2$, $\mathbb{COV}(I_{b_1 i}[t_1], I_{b_2 i}[t_2]) = C_{in}p[t_1]p[t_2]\mathbb{VAR}[w_{ji}]$.*

*Proof.* Since $I_{bi}[t] = \sum_{j=1}^{C_{in}} s_{bj}[t]w_{ji}$,

$$\mathbb{COV}(I_{b_1 i_1}[t_1], I_{b_2 i_2}[t_2]) = \mathbb{COV}\left(\sum_{j=1}^{C_{in}} s_{b_1 j}[t_1]w_{ji_1}, \sum_{j=1}^{C_{in}} s_{b_2 j}[t_2]w_{ji_2}\right) \tag{20}$$

When $(b_1, t_1) \neq (b_2, t_2)$ and $i_1 \neq i_2$, the lemma is trivial since the entries in the summation are all uncorrelated. For the case when $(b_1, t_1) = (b_2, t_2)$, we have:

$$
\begin{aligned}
\mathbb{COV}(I_{bi_1}[t], I_{bi_2}[t]) &= \mathbb{E}\left[\left(\sum_{j=1}^{C_{in}} s_{bj}[t]w_{ji_1}\right)\left(\sum_{j=1}^{C_{in}} s_{bj}[t]w_{ji_2})\right)\right] - \mathbb{E}[I_{bi_1}[t]]\mathbb{E}[I_{bi_2}[t]] \\
&= \sum_{j=1}^{C_{in}}\left(\mathbb{E}(s_{bj}[t]^2 w_{ji_1}w_{ji_2}) - \mathbb{E}(s_{bj}[t]w_{ji_1})\mathbb{E}(s_{bj}[t]w_{ji_2})\right) \\
&= \sum_{j=1}^{C_{in}}\left(p[t]\mathbb{E}^2[w_{ji_1}] - p[t]^2\mathbb{E}^2[w_{ji_1}]\right) \\
&= C_{in}\mathbb{E}^2[w_{ji}](p[t] - p[t]^2)
\end{aligned}
\tag{21}
$$

For the case when $i_1 = i_2$, we have:

$$
\begin{aligned}
\mathbb{COV}(I_{b_1i}[t_1], I_{b_2i}[t_2]) &= \mathbb{E}\left[\left(\sum_{j=1}^{C_{in}} s_{b_1j}[t_1]w_{ji}\right)\left(\sum_{j=1}^{C_{in}} s_{b_2j}[t_2]w_{ji})\right)\right] - \mathbb{E}[I_{b_1i}[t_1]]\mathbb{E}[I_{b_2i}[t_2]] \\
&= \sum_{j=1}^{C_{in}}\left(\mathbb{E}(s_{b_1j}[t_1]s_{b_2j}[t_2]w_{ji}^2) - \mathbb{E}(s_{b_1j}[t_1]w_{ji})\mathbb{E}(s_{b_2j}[t_2]w_{ji})\right) \\
&= \sum_{j=1}^{C_{in}}\left(p[t]^2\mathbb{E}[w_{ji}^2] - p[t_1]p[t_2]\mathbb{E}^2[w_{ji}]\right) \\
&= C_{in}p[t_1]p[t_2]\mathbb{VAR}[w_{ji}]
\end{aligned}
\tag{22}
$$

$\square$

After getting the mean, variance, and covariance of $I_{bi}[t]$, we can prove the following theorems by calculating the coefficient before the variance of $p[t]$:

**Theorem A.4.** *When Assumption 4.1 holds and the gross firing rate $p$ holds constant, then the expectation of sample variance of $\mu[t]$ among time-steps $\mathbb{E}\left[\frac{1}{T}\sum_{t=1}^{T}\mu[t]^2 - \mu^2\right]$ increases when the variance of firing rate among time-steps $\frac{1}{T}\sum_{t=1}^{T}p[t]^2 - p^2$ increases.*

*Proof.* Here we omit the subscript $b$ (the batch dimension) for $I_{bi}[t]$. First we calculate the expectation of $\mu[t]$ and $\mu$:

$$
\mathbb{E}[\mu[t]] = \mathbb{E}[I_{bi}[t]] = C_{in}p[t]\mathbb{E}[w_{ji}]
\tag{23}
$$

$$
\mathbb{E}[\mu] = \frac{1}{T}\sum_{t=1}^{T}\mathbb{E}[\mu[t]] = C_{in}p\mathbb{E}[w_{ji}]
\tag{24}
$$

Then we calculate the second moment, including $\mathbb{E}[\mu[t]^2]$ and $\mathbb{E}[\mu[t_1]\mu[t_2]]$:

$$\mathbb{E}[\mu[t]^2] = \mathbb{E}^2[\mu[t]] + \mathbb{VAR}[\mu[t]] = C_{in}^2 p[t]^2 \mathbb{E}^2[w_{ji}] + \frac{1}{C_{out}^2}\mathbb{VAR}\left(\sum_{i=1}^{C_{out}} I_i[t]\right)$$

$$=C_{in}^2 p[t]^2 \mathbb{E}^2[w_{ji}] + \frac{1}{C_{out}^2}\left(\sum_{i=1}^{C_{out}}\mathbb{VAR}(I_i[t]) + 2\sum_{1\leq i_1 < i_2 \leq C_{out}}\mathbb{COV}(I_{i1}[t], I_{i2}[t])\right)$$

$$=C_{in}^2 p[t]^2 \mathbb{E}^2[w_{ji}] + \frac{1}{C_{out}^2}\left(C_{out}C_{in}(p[t]\mathbb{E}[w_{ji}^2] - p[t]^2\mathbb{E}^2[w_{ji}]) + C_{out}(C_{out}-1)C_{in}\mathbb{E}^2[w_{ji}](p[t]-p[t]^2)\right)$$

$$=C_{in}^2 p[t]^2 \mathbb{E}^2[w_{ji}] + \frac{C_{in}}{C_{out}}\left(p[t]\mathbb{VAR}[w_{ji}] + C_{out}(p[t]-p[t]^2)\mathbb{E}^2[w_{ji}]\right). \tag{25}$$

$$\mathbb{E}[\mu[t_1]\mu[t_2]] = \mathbb{E}[\mu[t_1]]\mathbb{E}[\mu[t_2]] + \mathbb{COV}[\mu[t_1], \mu[t_2]]$$

$$=C_{in}^2 p[t_1]p[t_2]\mathbb{E}^2[w_{ji}] + \frac{1}{C_{out}^2}\mathbb{COV}\left(\sum_{i=1}^{C_{out}} I_i[t_1], \sum_{i=1}^{C_{out}} I_i[t_2]\right)$$

$$=C_{in}^2 p[t_1]p[t_2]\mathbb{E}^2[w_{ji}] + \frac{1}{C_{out}^2}\sum_{i=1}^{C_{out}}\mathbb{COV}\left(I_i[t_1], I_i[t_2]\right)$$

$$=C_{in}^2 p[t_1]p[t_2]\mathbb{E}^2[w_{ji}] + \frac{C_{in}}{C_{out}}p[t_1]p[t_2]\mathbb{VAR}[w_{ji}]. \tag{26}$$

Finally, we can calculate the target function:

$$\mathbb{E}\left[\frac{1}{T}\sum_{t=1}^{T}\mu[t]^2 - \mu^2\right] = \mathbb{E}\left[\frac{1}{T}\sum_{t=1}^{T}\mu[t]^2 - \left(\frac{1}{T}\sum_{t=1}^{T}\mu[t]\right)^2\right]$$

$$=\frac{T-1}{T^2}\sum_{t=1}^{T}\mathbb{E}[\mu[t]^2] - \frac{2}{T^2}\sum_{1\leq t_1 < t_2 \leq T}\mathbb{E}[\mu[t_1]\mu[t_2]]$$

$$=\frac{T-1}{T^2}\sum_{t=1}^{T}\left(C_{in}^2 p[t]^2\mathbb{E}^2[w_{ji}] + \frac{C_{in}}{C_{out}}\left(p[t]\mathbb{VAR}[w_{ji}] + C_{out}(p[t]-p[t]^2)\mathbb{E}^2[w_{ji}]\right)\right)$$

$$- \frac{2}{T^2}\sum_{1\leq t_1 < t_2 \leq T}\left(C_{in}^2 p[t_1]p[t_2]\mathbb{E}^2[w_{ji}] + \frac{C_{in}}{C_{out}}p[t_1]p[t_2]\mathbb{VAR}[w_{ji}]\right)$$

$$=C_{in}^2\mathbb{E}^2[w_{ji}]\left(\frac{1}{T}\sum_{t=1}^{T}p[t]^2 - p^2\right) + C_{in}\mathbb{E}^2[w_{ji}]\left(\frac{T-1}{T}p - \frac{T-1}{T^2}\sum_{t=1}^{T}p[t]^2\right)$$

$$+ \frac{C_{in}}{C_{out}}\mathbb{VAR}[w_{ji}]\left(\frac{T-1}{T}p + \frac{1}{T^2}\sum_{t=1}^{T}p[t]^2 - p^2\right) \tag{27}$$

When $p$ is constant, the variance among time-steps only depends on $\sum_{t=1}^{T}p[t]^2$. In the last equation of Eq. 27, the only thing that can vary is $\sum_{t=1}^{T}p[t]^2$, and the coefficient in front of it is always positive ($C_{in}^2\mathbb{E}^2[w_{ji}]\cdot\frac{1}{T} \geq C_{in}\mathbb{E}^2[w_{ji}]\cdot\frac{1}{T} \geq C_{in}\mathbb{E}^2[w_{ji}]\cdot\frac{T-1}{T^2}$). Therefore, the conclusion holds. $\square$

**Theorem A.5.** *When Assumption 4.1 holds and the gross firing rate $p$ keeps constant, then the expectation of variance within time-steps $\mathbb{E}[\frac{1}{T}\sum_{t=1}^{T}\sigma^2[t]]$ keeps constant.*

*Proof.* We first calculate $\mathbb{E}[\sigma^2[t]]$ and then sum them up. The $\mathbb{E}[\sigma^2[t]]$ can be split into calculating $\mathbb{E}[I_i[t]^2]$ and $\mathbb{E}[\mu[t]^2]$, which have been calculated before.

$$
\mathbb{E}[\sigma^2[t]] = \mathbb{E}\left[\frac{1}{C_{out}}\sum_{i=1}^{C_{out}} I_i[t]^2 - \mu[t]^2\right] = \frac{1}{C_{out}}\sum_{i=1}^{C_{out}}\mathbb{E}[I_i[t]^2] - \mathbb{E}[\mu[t]^2]
$$

$$
= C_{in}p[t]\mathbb{E}[w_{ji}^2] + C_{in}(C_{in}-1)p[t]^2\mathbb{E}^2[w_{ji}] - C_{in}^2 p[t]^2\mathbb{E}^2[w_{ji}]
$$

$$
-\frac{C_{in}}{C_{out}}\left(p[t]\mathbb{VAR}[w_{ji}] + C_{out}(p[t]-p[t]^2)\mathbb{E}^2[w_{ji}]\right)
$$

$$
= C_{in}p[t]\mathbb{E}[w_{ji}^2] - C_{in}p[t]^2\mathbb{E}^2[w_{ji}] - \frac{C_{in}}{C_{out}}\left(p[t]\mathbb{VAR}[w_{ji}] + C_{out}(p[t]-p[t]^2)\mathbb{E}^2[w_{ji}]\right)
$$

$$
= C_{in}(p[t]\mathbb{VAR}[w_{ji}] + (p[t]-p[t]^2)\mathbb{E}^2[w_{ji}]) - \frac{C_{in}}{C_{out}}\left(p[t]\mathbb{VAR}[w_{ji}] + C_{out}(p[t]-p[t]^2)\mathbb{E}^2[w_{ji}]\right)
$$

$$
= \frac{C_{in}(C_{out}-1)}{C_{out}}p[t]\mathbb{VAR}[w_{ji}] \tag{28}
$$

$$
\mathbb{E}\left[\frac{1}{T}\sum_{t=1}^{T}\sigma^2[t]\right] = \frac{1}{T}\sum_{t=1}^{T}\frac{C_{in}(C_{out}-1)}{C_{out}}p[t]\mathbb{VAR}[w_{ji}] = \frac{C_{in}(C_{out}-1)}{C_{out}}\cdot p\cdot\mathbb{VAR}[w_{ji}] \tag{29}
$$

As a result, the variance of $p[t]$ will not affect $\mathbb{E}\left[\frac{1}{T}\sum_{t=1}^{T}\sigma^2[t]\right]$, which means it keeps constant. $\qquad\square$

## B  ALGORITHM DESCRIPTION FOR OUR METHOD

Our algorithm works under the online learning framework, which means the network goes through forward and backward propagations step by step from time-step 1 to $T$ (instead of first forward from time step 1 to $T$ and then backward from time step $T$ to 1). Since the network is processed step by step, it does not require saving the intermediate state from time-step 1 to $T$ as in regular BPTT. In each time-step, the information goes from the input layer to the output layer of the network in the forward pass, and then the gradients go from the output layer to the input layer in the backward pass.

**The workflow of each layer** is shown in Algorithm 1, while **the calculation of** $\mu[t], \sigma^2[t], \hat{\mu}, \hat{\sigma^2}$ is shown separately in Algorithm 2:

## C  IMPLEMENTATION DETAILS

We conduct experiments on CIFAR10, CIFAR100, DVS-Gesture, CIFAR10DVS, and Imagenet datasets. The network structure of VGGSNN we use for the CIFAR10, CIFAR100, DVS-Gesture, CIFAR10-DVS datasets is consistent with OTTT (64C3-128C3-AP2-256C3-256C3-AP2-512C3-512C3-AP2-512C3-512C3-GAP-FC), where 64C3 denotes convolution layer with $3 \times 3$ convolution kernel and 64 output channels, AP2 means $2 \times 2$ average pooling, GAP means global average pooling, and FC means fully connected layer. For Imagenet classification, we just use standard Resnet-34 architecture.

In all experiments, we use an SGD optimizer with a momentum of 0.9 with a cosine annealing learning rate scheduler. The data augmentation we use for each dataset is listed as follows: For CIFAR10 and

---

**Algorithm 1** The workflow of each layer

---

**Input:** Output of the last layer $\boldsymbol{s}^{l-1}[t]$ (input spike train/image at time $t$ for the input layer) and the weight between last layer and current layer $\boldsymbol{W}^l$ ($\boldsymbol{s}^{l-1}[t]$ and $\boldsymbol{W}^l$ are both tensors instead of scalars).

// 1. Calculate input current $\boldsymbol{I}[t]$

$\boldsymbol{I}[t] = layer(\boldsymbol{s}^{l-1}[t], \boldsymbol{W}^l)$           $\triangleright$ $layer$ means a conv layer, a linear layer or other types of layer

// 2. Apply OSR on $\boldsymbol{I}[t]$ to get the normalized $\tilde{I}[t]$

**if** training **then**

    Calculate $\mu[t], \sigma^2[t], \hat{\mu}, \hat{\sigma^2}$ according to Algorithm. 2

    $\hat{I}[t] = \frac{\boldsymbol{I}[t] - \mu[t]}{\sqrt{\sigma^2[t]+\epsilon}}$

    $\tilde{I}[t] = \gamma \cdot \left( \hat{I}[t] \cdot no\_grad\left(\frac{\sqrt{\sigma^2[t]+\epsilon}}{\sqrt{\hat{\sigma^2}+\epsilon}}\right) + no\_grad\left(\frac{\mu[t]-\hat{\mu}}{\sqrt{\hat{\sigma^2}+\epsilon}}\right) \right) + \beta$

**else**

    $\tilde{I}[t] = \gamma \cdot \frac{\boldsymbol{I}[t]-\hat{\mu}}{\sqrt{\hat{\sigma^2}+\epsilon}} + \beta$     $\triangleright$ Note that same linear transformations are applied in training and inference

**end if**

// 3. Update membrane potential of neurons in layer $l$ according to the LIF neuron model and input $\tilde{I}[t]$

$\boldsymbol{u}^l[t] = (1 - \frac{1}{\tau^l})\boldsymbol{u}^l[t-0.5] + \tilde{I}[t]$

// 4. Apply OTS to update the threshold $\theta[t]$

$\theta[t] = \mu_{\mathrm{mem}}[t] + \sigma_{\mathrm{mem}}[t] \cdot \frac{\theta[1]-\mu_{\mathrm{mem}}[1]}{\sigma_{\mathrm{mem}}[1]}$

// 5. Fire spikes $\boldsymbol{s}^l[t]$ and then reset membrane potential

$\boldsymbol{s}^l[t] = \Theta(\boldsymbol{u}^l[t] - \theta[t])$

$\boldsymbol{u}^l[t+0.5] = \boldsymbol{u}^l[t](1 - \boldsymbol{s}^l[t])$

---

**Algorithm 2** The calculation of $\mu[t], \sigma^2[t], \hat{\mu}, \hat{\sigma^2}$

---

**Input:** (Additional Input) Current time-step $t$ ($1 \leq t \leq T$)     $\triangleright$ To determine whether we should initialize variables or calculate running mean/variance at the current time-step

**Output:** $\nabla \boldsymbol{W}^{(n)}(n = 1, ..., N)$

// 1. Calculate the batch mean $\mu[t]$ and variance $\sigma^2[t]$ according to $\boldsymbol{I}[t]$ according to Eq. 13 (Here $m$ is the number of elements in a channel, which forms a group for normalization).

$\mu[t] = \frac{1}{m}\sum_{x=1}^{m} I_x[t]$

$\sigma^2[t] = \frac{1}{m}\sum_{x=1}^{m}(I_x[t] - \mu[t])^2$

// 2. According to Eq. 14 15, we need variables $\mu$ and $\sigma^2$ to accumulate total mean and variance. Before the first time-step, we initialize $\mu$ and $\sigma^2$ to 0:

**if** $t = 1$ **then**

    $\mu \leftarrow 0, \sigma^2 \leftarrow 0$

**end if**

// Then we accumulate total mean and variance according to Eq. 14 15:

$\mu \leftarrow \mu + \frac{1}{T}\mu[t], \sigma^2 \leftarrow \sigma^2 + \frac{1}{T}(\sigma^2[t] + \mu[t]^2)$

// 3. Calculate running mean $\hat{\mu}$ and running variance $\hat{\sigma^2}$ in time-step $T$ (the last time step)

**if** $t = T$ **then**

    $\sigma^2 \leftarrow \sigma^2 - \mu^2$

    $\hat{\mu} \leftarrow \hat{\mu} + (1 - momentum)(\mu - \hat{\mu})$           $\triangleright$ We take $momentum = 0.9$ as in BN.

    $\hat{\sigma^2} \leftarrow \hat{\sigma^2} + (1 - momentum)(\sigma^2 - \hat{\sigma^2})$

**end if**

---

Table 3: Experimental configurations

| Dataset | CIFAR10 | CIFAR100 | DVS-Gesture | CIFAR10-DVS | Imagenet |
|---|---|---|---|---|---|
| Epochs | 300 | 300 | 300 | 300 | 100 |
| Batch size | 128 | 128 | 128 | 128 | 256 |
| Learning rate | 0.1 | 0.1 | 0.01 | 0.1 | 0.1 |
| Weight decay | 5e-4 | 5e-4 | 5e-4 | 5e-4 | 2e-5 |
| MSE weight $\epsilon$ | 0.05 | 0.05 | 0.001 | 0.001 | 0.05 |
| Dropout rate | 0 | 0 | 0.05 | 0.1 | 0 |

CIFAR100, we use RandomCrop(4) + Cutout() + RandomHorizontalFlip() + Normalize(); For DVS-Gesture, we use RandomResizedCrop(128, scale=(0.7, 1.0)) + Resize(48) + RandomRotation(20) + RandomTemporalDelete(14) (recall the total time-step is 20, and the random temporal delete drops 6 time-steps (30%)). For CIFAR10-DVS, we use the neuromorphic data augmentation (NDA) which comes from (Li et al., 2022). For Imagenet, we use RandomResizedCrop(224) + RandomHorizontalFlip() + Normalize() during training. During testing, the image is first resized to $256 \times 256$ and center-cropped to $224 \times 224$ and then normalized. Other hyperparameters we use are provided in Table 3, including total training epochs, batch size, learning rate, weight decay, $\epsilon$ (weight of MSE loss in Eq. 5), and dropout rate.

For the configuration of **Vanilla BN** used as the baseline, it calculates the statistics solely based on data from each time step. In this approach, at every time step during training, the normalized input $\tilde{I}[t]$ is computed by

$$\tilde{I}[t] = \gamma \cdot \frac{I[t] - \mu[t]}{\sqrt{\sigma^2[t] + \epsilon}} + \beta,$$

where the batch mean $\mu[t]$ and batch variance $\sigma^2[t]$ at time-step $t$ accords with Algorithm 2 in the above.

The slight difference between it and the standard BN is the calculation of running mean and variance: If we use the same momentum as in our OSR to update running mean and variance at each time step, they will be unstable since they are updated $T$ times more compared with our OSR. A simple approach is to change the $momentum$ parameter to $1 - (1 - momentum)/T$, but we choose to implement it in a strict corresponding way: We accumulate the mean and variance during training (for the variance, we accumulate by $\sigma^2 \leftarrow \sigma^2 + \frac{1}{T}\sigma^2[t]$ instead of the complex way shown above), and only update the running mean and variance at time step $T$. In this way, there is no need to change the $momentum$ parameter.

## D ABLATION STUDY

Here we show the ablation results of our proposed modules: online spiking renormalization (OSR) and online threshold stabilizer (OTS). Since the OTS is proposed to help the training of OSR, we provide the results of Vanilla BN / OSR / OSR+OTS here on CIFAR10/100 and Imagenet dataset. The results are shown in Table. 4. It is shown that adding OSR will improve the performance over vanilla BN, and adding OTS will further improve the performance over solely adding OSR. It is worth noting that only adding OSR is sensitive to the $weight\ decay$ parameter, it often requires lower $weight\ decay$ to get a better result. For example, both Resnet-19+(Vanilla BN) and Resnet-19+OSR+OTS can be trained on CIFAR10 with a weight decay of 2e-4 while Resnet-19+OSR cannot. Another example is that the performance of SEW-Resnet-34+OSR will degrade to 54.97% on Imagenet when using a weight decay of 2e-5. On the other hand, OSR+OTS is much more stable with large weight decay parameters.

Table 4: Ablation results

|  | CIFAR10 Acc (wd)* | CIFAR100 Acc (wd) | Imagenet Acc (wd) |
|---|---|---|---|
| VGG+Vanilla BN | 92.6 (5e-4) | 75.17 (5e-4) | - |
| VGG+OSR | 94.05 (2e-4) | 75.65 (2e-4) | - |
| VGG+OSR+OTS | 94.35 (5e-4) | 76.48 (5e-4) | - |
| Resnet-19+Vanilla BN | 92.96 (2e-5) | 73.68 (2e-4) | - |
| Resnet-19+OSR | 95.14 (2e-5) | 74.03 (2e-5) | - |
| Resnet-19+OSR+OTS | 95.20 (2e-5) | 77.86 (2e-4) | - |
| SEW-Resnet-34+Vanilla BN | - | - | 60.48 (2e-5) |
| SEW-Resnet-34+OSR | - | - | 61.92 (0) |
| SEW-Resnet-34+OSR+OTS | - | - | 64.14 (2e-5) |

\* We report both accuracy and weight decay statistics here.

## E  ADDITIONAL EXPERIMENTS

**Necessity of "double transformation" in OSR.** The OSR mechanism is shown to be useful among many mechanisms that we have tried. One simpler mechanism that does not work well is directly using a "linear transformation" instead of the "double transformation" in OSR. it directly applies

$$\tilde{I}[t] = \gamma \cdot \frac{I[t] - \hat{\mu}}{\sqrt{\hat{\sigma^2} + \epsilon}} + \beta$$

in both training and inference. We have tested this approach on the CIFAR100 dataset using VGGSNN and find it very hard to train. The final result we get is 53.25%, which is significantly worse than OSR.

**Fixed $\theta[t]$ during inference in OTS.** In OTS, the threshold $\theta[t]$ is dynamically adjusted for each sample batch during both the training and inference phases. It will be better when $\theta[t]$ is fixed during the inference stage if there is no significant performance decrease since inference batch size will not affect performance and it is more friendly to neuromorphic chips under this case. Hence we have conducted two extra experiments:

1. We have tested the performance of using fixed running $\theta[t]$ on Imagenet (using our saved model), its performance is 64.06% (original accuracy is 64.14%) (using the saved model of OSR+OTS). This result shows that fixed $\theta$ works well.

2. We have tested the performance for $batchsize = 1$ on Imagenet (also using our saved model), and the performance is 62.64%. Although there is a performance drop, it is still better than the baseline.

## F  MEMBRANE POTENTIAL VISUALIZATION

To see whether the Gaussian assumption in OTS is reasonable, we collect the membrane potential of a VGG network trained with OSR and OTS on the CIFAR-10 dataset and visualize the distribution of membrane potentials for each layer and each time step. The results are shown in Fig. 3. These distributions display the shape of bell curves, which indicate the similarity between these distributions and Gaussian distributions. Most of the distributions take the mean value around zero. This result shows that the Gaussian assumption in OTS is reasonable. Therefore, our proposed algorithm exploits the adaptation of batch normalization and can cope with the varied distributions of network features during online learning of spiking neural networks.

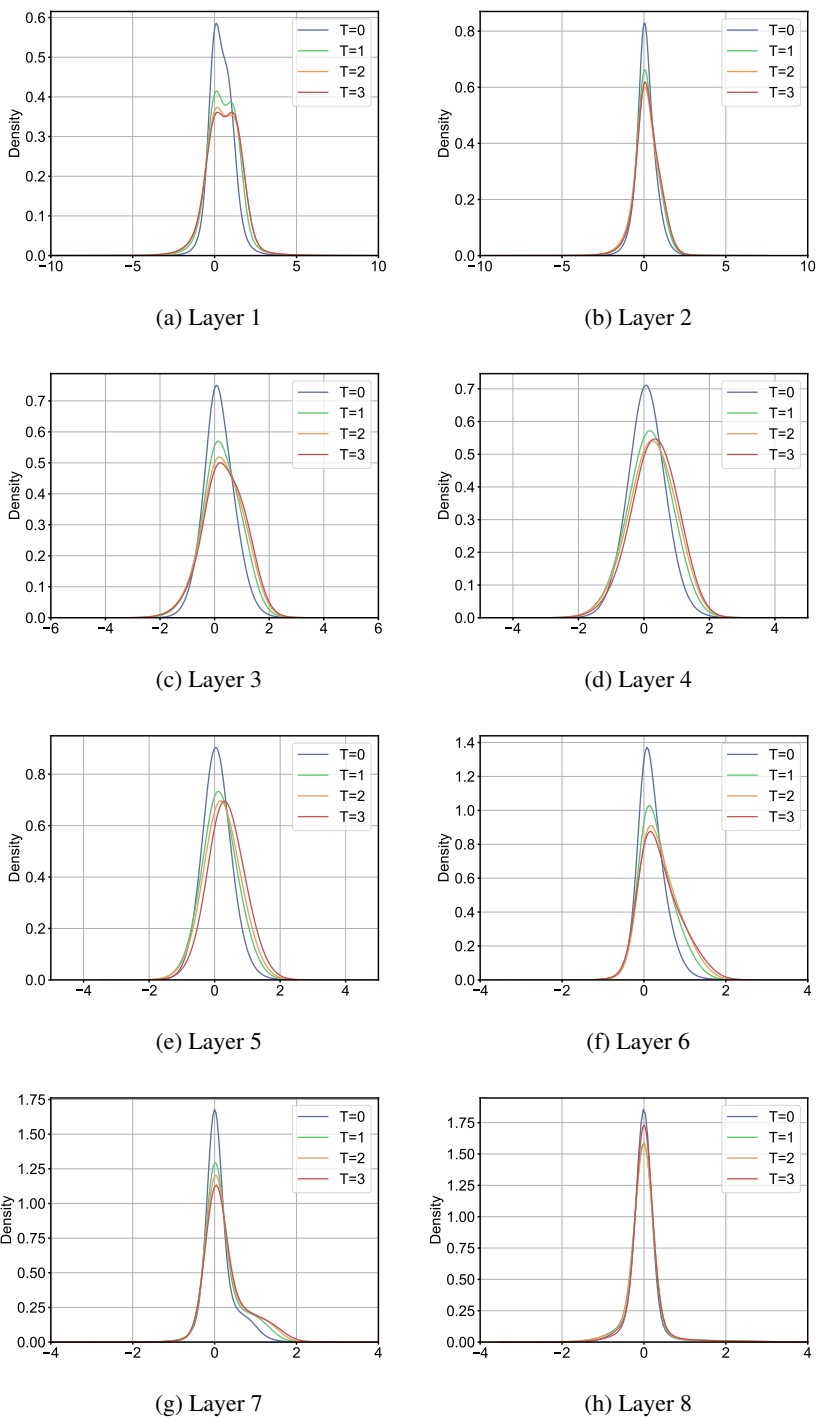

Figure 3: Visualization of the distributions of membrane potentials for each layer and each time step.

