# OpenReview forum: "Online Stabilization of Spiking Neural Networks"
_ICLR.cc/2024/Conference — ICLR 2024 spotlight_

### Official Review · Reviewer_W97A · 2023-10-27

**Soundness:** 2 fair
**Presentation:** 2 fair
**Contribution:** 2 fair
**Rating:** 6
**Confidence:** 5

**Summary:**

When batch normalization (BN) is implemented in spiking neural network (SNN) structures, it is a common practice to compute the statistics across all time steps for running BN. However, this commonly used BN is ill-suited for the online training of SNNs. To enable BN for online training, this paper introduces two strategies called Online Spiking Renormalization (OSR) and Online Threshold Stabilizer (OTS), which preserve the online training property and memory efficiency. Experimental results demonstrate the efficacy of the proposed methods.

**Strengths:**

1. The proposed tricks are both simple to implement and logically sound, contributing to their practicality and ease of use.
2. The performance is good when compared to online training methods and conventional methods.

**Weaknesses:**

1. If the reviewer understands the OTS method correctly, the threshold $\theta[t]$ is dynamically adjusted for each sample batch during both the training and inference phases. However, the reviewer thinks that $\theta[t]$ should be precalculated and fixed during the inference stage. Otherwise, the batch size will highly influence the performance. Much worsely, the obtained network cannot be implemented on normal neuromorphic chips because the chips do not support calculating the mean and variance. the reviewer suggests the authors to maintain a running $\theta[t]$ used for inference.


2. This paper could be regarded as an engineering work. Then the reviewer thinks that the ablation experiments are not abundant:
 (i). When OSR is incorporated individually, there appears to be a significant performance degradation. It is important to clarify whether such a phenomenon is typical across various datasets.
 (ii). A simpler approach to integrating BN into the online training regime involves calculating the statistics solely based on data from each time step. In this approach, at every time step during training, the normalized I[t] is computed based on $\mu[t]$ and $\sigma[t]$, as illustrated in eq. 8. Additionally, the running mean and variance are also updated at each time step. This vanilla BN method should be considered as the baseline. It would be valuable to assess whether this method outperforms OSR and whether the combination of the baseline with OTS outperforms the combination of OSR with OTS.
 (iii). In the backward stage, the authors utilize a "double transformation" trick to facilitate more meaningful backpropagation. What if we directly conduct bachprop based on the ``linear transformation''? Is OSR better than that?


Minor:
1. eq.2: $s^{l-1}[t]W^l$ -> $W^ls^{l-1}[t]$, eq.4: $u^{l} [t] (1-s^{l} [t] )$  -> $u^{l}[t] \odot (1-s^{l}[t])$
2. The ``Online Calculation of All-time Mean and Variance'' part appears somewhat trivial. The reviewer thinks that it might be unnecessary to include this information in the list of contributions. The audiences may expect a more significant contribution when such a detail is highlighted.
3. The statement preceding Section 5: ``our OTS mechanism helps our OTS mechanism''.

**Questions:**

1. The line right before Section 4.3: is there typos in the formula? Should the formula be $...(\theta[1]-\mu[1])/\sigma[1]$? Then does it mean that $\theta[1]$ is fixed all the time?
2. How to implement the proposed method on NF-Resnet-34 which is a normalizer-free architecture? The reviewer knows that the goal of this experiment is to compare the proposed method and OTTT. But still, the proposed method and normalizer-free nets are totally orthogonal.

---

> ### Author Response · Authors · 2023-11-21
> **Response to Reviewer W97A**
>
> We sincerely appreciate your thoughtful and comprehensive review. Your insights highlight crucial aspects we overlooked or omitted in our paper. Each point you raised significantly contributes to enhancing the quality of our work. We are dedicated to addressing your concerns comprehensively and offering detailed responses in the upcoming sections.
>
>
> > If the reviewer understands the OTS method correctly, the threshold $\theta[t]$ is dynamically adjusted for each sample batch during both the training and inference phases. However, the reviewer thinks that $\theta[t]$ should be precalculated and fixed during the inference stage. Otherwise, the batch size will highly influence the performance. Much worsely, the obtained network cannot be implemented on normal neuromorphic chips because the chips do not support calculating the mean and variance. the reviewer suggests the authors to maintain a running $\theta[t]$ used for inference.
>
> Thank you for your valuable suggestion. Your understanding is right that the threshold $\theta[t]$ is dynamically adjusted for each sample batch during both the training and inference phases.
> We have conducted two extra experiments:
> - 1. We have tested the performance of using fixed running $\theta[t]$ on Imagenet, its performance is 64.06\% (original accuracy is 64.14\%) (using the saved model of OSR+OTS). This result shows that your suggestion works well.
> - 2. We have tested the performance for $batch size=1$ on Imagenet (using our saved model), and the performance is 62.64\%. Although there is a performance drop, it is still better than the baseline.
>
> In addition, we think that $\theta[t]$ could be dynamical during the inference stage if it is incorporated in the neuron model, since it can be viewed as a neural adaptation mechanism. This is a potential direction for future research.
>
>
> > This paper could be regarded as an engineering work. Then the reviewer thinks that the ablation experiments are not abundant:
>     > (i). When OSR is incorporated individually, there appears to be a significant performance degradation. It is important to clarify whether such a phenomenon is typical across various datasets.
>     > (ii). A simpler approach to integrating BN into the online training regime involves calculating the statistics solely based on data from each time step. In this approach, at every time step during training, the normalized $I[t]$ is computed based on $\mu[t]$ and $\sigma[t]$, as illustrated in eq. 8. Additionally, the running mean and variance are also updated at each time step. This vanilla BN method should be considered as the baseline. It would be valuable to assess whether this method outperforms OSR and whether the combination of the baseline with OTS outperforms the combination of OSR with OTS.
>     > (iii). In the backward stage, the authors utilize a "double transformation" trick to facilitate more meaningful backpropagation. What if we directly conduct backprop based on the ''linear transformation''? Is OSR better than that?
>
> Thank you for mentioning this crucial point. Actually, the experimental baseline (BN(Vanilla) for the Imagenet dataset in Table 1) in the paper is almost the same (with slight difference) as you mentioned in (ii). The details are provided in **Experimental settings** in **Response to All Reviewers**.
> - For (i), We have conducted extra experiments on CIFAR10 and CIFAR100 dataset, and the results are shown in the following table:
>
> |                     | CIFAR10 | CIFAR100 |
> | ------------------- | ------- | -------- |
> | VGG+Vanilla BN      | 92.6    | 75.17    |
> | VGG+OSR             | 94.05   | 75.65    |
> | VGG+OSR+OTS         | 94.35   | 76.48    |
> | Resnet19+Vanilla BN | 92.96   | 73.68    |
> | Resnet19+OSR        | 95.14   | 74.03    |
> | Resnet19+OSR+OTS    | 95.20   | 77.86    |
>
> Note that Resnet19+OSR on CIFAR100 has a significant performance drop compared with Resnet19+OSR+OTS.
> In addition, when conducting these experiments, we find that only adding OSR is sensitive to the $weight$ $decay$ parameter. It often requires lower $weight$ $decay$ to get a better result. The ablation experiments we reported in Table 1 uses the same set of parameters for vanilla BN, BN+OSR, BN+OTS, and BN+OSR+OTS. Maybe turning down the $weight$ $decay$ parameter for BN+OSR will promote its performance, but we do not have enough time to finish tuning this parameter during rebuttal. We are working on this and will provide a better result if we can.
>
> - For (iii), we have tested this ''linear transformation'' on the CIFAR100 dataset using VGGSNN and find it very hard to train. The final result we get is 53.25\%, which is significantly worse than OSR.

---

> ### Author Response · Authors · 2023-11-21
> **Response to Reviewer W97A (part 2)**
>
> > eq.2: $s^{l-1}[t] W^l \rightarrow W^l s^{l-1}[t]$, eq.4: $u^{l}[t] (1 - s^{l}[t]) \rightarrow u^{l}[t] \odot (1 - s^{l}[t])$
>
> Thank you for such a careful review. We have revised these equations in the latest version.
>
>
> > The ``Online Calculation of All-time Mean and Variance'' part appears somewhat trivial. The reviewer thinks that it might be unnecessary to include this information in the list of contributions. The audiences may expect a more significant contribution when such a detail is highlighted.
>
> Thank you for your suggestion. We have combined this part into the contribution of OSR.
>
>
> > The statement preceding Section 5: ''our OTS mechanism helps our OTS mechanism''.
>
> Thank you for the careful review. We have changed it to ''our OTS mechanism helps our OSR mechanism'' in the latest version.
>
>
> > The line right before Section 4.3: is there typos in the formula? Should the formula be $\theta[t] = \mu_\text{mem}[t] + \sigma_\text{mem}[t] \cdot \frac{\theta[1] - \mu_\text{mem}[1]}{\sigma_\text{mem}[1]}$? Then does it mean that $\theta[1]$ is fixed all the time?
>
> I appreciate your thorough review, and thank you for bringing up this specific point. You are right that the formula should be $\theta[t] = \mu_\text{mem}[t] + \sigma_\text{mem}[t] \cdot \frac{\theta[1] - \mu_\text{mem}[1]}{\sigma_\text{mem}[1]}$, and we have fixed it in the latest version. We set $\theta[1] = 1$ for all training iterations.
>
>
> > How to implement the proposed method on NF-Resnet-34 which is a normalizer-free architecture? The reviewer knows that the goal of this experiment is to compare the proposed method and OTTT. But still, the proposed method and normalizer-free nets are totally orthogonal.
>
> Thank you for bringing up this point. The NF-Resnet-34 in OTTT uses a normalizer-free network adding membrane potential in the shortcut connection.
> Briefly speaking, we keep adding membrane potential in the shortcut connection while replacing the normalizer-free part with our normalization modules.
> Specifically, we do not use scaled weight standardization, and drop scaling factors $\alpha, \beta$ along with the corresponding operations to keep variance stable. Instead, we use our normalization modules here.

---

> ### Comment · Reviewer_W97A · 2023-11-23
>
> Thanks for the clear rebuttal. Now my concerns have been addressed and I am glad to increase my score from 5 to 6. I suggest the authors to revise the manuscript accordingly.
>
> By the way, I am interested in how $\theta[t]$ could be dynamical during the inference stage when implemented on neuromorphic hardware. How could it be incorporated in an adaptive LIF model since it does not follow some explicit dynamics?

---

### Official Review · Reviewer_pFYW · 2023-11-01

**Soundness:** 3 good
**Presentation:** 3 good
**Contribution:** 3 good
**Rating:** 8
**Confidence:** 5

**Summary:**

This paper focuses on the online training of SNNs. It adds the normalization mechanism into online training, which have not yet been fully explored by previous works. It proposes two modules to improve the standard Batch Normalization, named Online Spiking Renormalization and Online Threshold Stabilizer, which ensure consistent parameters and stable neuron firing rates across time steps. The paper demonstrates the effectiveness of the proposed methods on various datasets and shows that they outperform existing state-of-the-art online training methods.

**Strengths:**

1. The paper presents a novel approach to training spiking neural networks that integrates essential batch normalization into the online training process by introducing online spiking renormalization and online threshold stabilizers to enhance training stability.

2. The paper is well-organized and well-written. Also, the figures and tables are well-designed and provide a clear representation of the results.

3. The proposed method outperforms existing state-of-the-art online training methods.

**Weaknesses:**

1. Although the online training approaches save the memory cost, the proposed method falls short of BPTT in performance.

**Questions:**

1. What is the difference between OSR and batch renormalization [1] ?

2. The online calculation of all-time mean and variance is interesting. However, where is it used? Is it a part of OSR?

3. The authors have made an assumption that the membrane potential follows a normal distribution in OTS. Although experiments have shown the effectiveness of OTS+OSR, I am still curious about the real distribution of the membrane potential.

[1]Sergey Ioffe. Batch renormalization: Towards reducing minibatch dependence in batch-normalized models. Advances in neural information processing systems, 30, 2017

I consider increasing my score if the authors can solve my concerns.

---

> ### Author Response · Authors · 2023-11-21
> **Response to Reviewer pFYW**
>
> We deeply appreciate your valuable feedback and acknowledgment of our method's novelty, the paper's organization, and writing quality. Your insights are invaluable. We are committed to thoroughly addressing your concerns and providing detailed responses, as outlined in the following sections.
>
> > Although the online training approaches save the memory cost, the proposed method falls short of BPTT in performance.
>
> Online training saves memory cost by backpropagating each time-step using information generated at the current time-step or at previous time-steps.
> This advantage is the direct origin of its disadvantage that it cannot use the information from future time-steps (compared with BPTT-based training algorithms).
> At the same time, it is hard to find a way to completely offset the information from future time-steps in online training, especially when the network has multiple layers (perfect solution such as eligibility traces [1][2] only works for single-layer networks without other mechanisms like BN).
> As a result, the performance of online training approaches still fall short of BPTT.
>
>     [1] Bellec, G., Scherr, F., Subramoney, A., Hajek, E., Salaj, D., Legenstein, R., & Maass, W. (2020). A solution to the learning dilemma for recurrent networks of spiking neurons. Nature communications, 11(1), 3625.
>     [2] Bohnstingl, T., Woźniak, S., Pantazi, A., & Eleftheriou, E. (2022). Online spatio-temporal learning in deep neural networks. IEEE Transactions on Neural Networks and Learning Systems.
>
>
> > What is the difference between OSR and batch renormalization [1] ?
>
> You have raised a critical point.
> Since our OSR is applied in spiking neural networks, it involves a time dimension, which is not encountered by batch renormalization.
> Actually, OSR exploits additional advantage by this additional time dimension:
> One key aim of OSR is to alleviate the temporal covariate shift by applying the same forward transformation to input currents at all time-steps.
> Although OSR achieves this goal by the same form of double transformation as batch renormalization, this problem that OSR solves is not the aim of batch renormalization due to the extra time dimension.
> In addition, to make the memory cost perfectly unrelated to the number of time-steps in simulation, OSR requires online calculation of all-time mean and variance.
>
>     [1]Sergey Ioffe. Batch renormalization: Towards reducing minibatch dependence in batch-normalized models. Advances in neural information processing systems, 30, 2017
>
>
> > The online calculation of all-time mean and variance is interesting. However, where is it used? Is it a part of OSR?
>
> Yes, it is a part of OSR.
> The process of OSR is described as follows:
>
> It is applied on $I[t]$ to get the normalized $\tilde{I}[t]$:.
>   During training, first we normalize $I[t]$ to $\hat{I}[t]$ (note $\hat{I}[t]$ passes gradients to $\mu[t]$ and $\sigma^2[t]$):
>   $$\hat{I}[t] = \frac{I[t] - \mu[t]}{\sqrt{\sigma^2[t] + \epsilon}}$$
> Then we apply linear transformation from $\hat{I}[t]$ to $\tilde{I}[t]$ (note $\tilde{I}[t]$ does not pass gradients to $\mu[t]$ and $\sigma^2[t]$ due to $no\\_grad()$, and $\gamma, \beta$ are learnable):
>   $$\tilde{I}[t] = \gamma \cdot \left( \hat{I}[t] \cdot no\\_grad\left( \frac{\sqrt{\sigma^2[t] + \epsilon}}{\sqrt{\hat{\sigma^2} + \epsilon}} \right) + no\\_grad\left(\frac{\mu[t] - \hat{\mu}}{\sqrt{\hat{\sigma^2} + \epsilon}} \right) \right) + \beta$$
>   As a result, $\tilde{I}[t] = \gamma \cdot \frac{ I[t] - \hat{\mu}}{\sqrt{\hat{\sigma^2} + \epsilon}} + \beta$ in the forward pass in training. Note that we apply conventional linear transformation $\tilde{I}[t] = \gamma \cdot \frac{ I[t] - \hat{\mu}}{\sqrt{\hat{\sigma^2} + \epsilon}} + \beta$ in the testing phase, so the input goes through the same linear transformations **among all time-steps in both training and inference**.
>
> The calculation of $\mu[t], \sigma^2[t], \hat{\mu}, \hat{\sigma^2}$ during training can be found in **Response to All Reviewers**.
>
>
> > The authors have made an assumption that the membrane potential follows a normal distribution in OTS. Although experiments have shown the effectiveness of OTS+OSR, I am still curious about the real distribution of the membrane potential.
>
> Thanks for your valuable advice. To verify the assumption that the membrane potential follows a normal distribution, we collect the membrane potential of a VGG network trained with OSR and OTS on the CIFAR-10 dataset and visualize the distribution of membrane potentials for each layer and each time step. We display the results in Fig. 3 in the appendix of the revised paper. These distributions displays the shape of bell curves, which similars to normal distributions. Thus, we think it is reasonable to use the assumption of normal distribution in OTS.

---

> > ### Comment · Reviewer_pFYW · 2023-11-22
> > **My concerns have been addressed**
> >
> > I have increased my score as the authors have solved all my concerns. It would be better if the authors can add these clarifications in the final version.

---

### Official Review · Reviewer_2ic1 · 2023-11-03

**Soundness:** 4 excellent
**Presentation:** 4 excellent
**Contribution:** 4 excellent
**Rating:** 8
**Confidence:** 4

**Summary:**

The paper considers the online setting of SNN, and points out one important mismatch of BN happens in the training and testing stages. The authors proposed one nice solution to solve the issue, and the experimental results support the benefits with the new algorithms. Importantly, in addition to the experimental verification, the authors also provide necessary theoretical analysis to the new algorithm.

**Strengths:**

1. The paper have a very clear motivation: mismatch of BN happens in the training and testing stages.
2. The authors proposed one nice solution to solve the issue, and the experimental results support the benefits with the new algorithms.
3. The propblem of this paper is unsolved in the comminity before. The solution of this paper is interesting and novel. Importantly, it signicantly improve the performance. Considering the important role of BN in ML, I think the signicance and novelty of this work is good.
4. The authors also provide the necessary theoretical analysis to the new algorithm.
5. Presentation is good. Especially, I like figure 1 to intutiively explain the proposed algorithm.

**Weaknesses:**

The paper is with a nice story, and I especially like the figure 1 to intutiively explain the proposed algorithm. However, I still have some questions or comments below.

1. What is the intuitive reason to have double normalization in (12)
2. I suggest to provide the experimental results to verify the Gaussion assumption.
3. I did not find the calculation form of $\hat{\mu}$ and $\\hat{\sigma}$. I suggest to provide a clear definition of them in the paper.

Minor issue:
1. I did not find  the meaning of m in (8).

**Questions:**

Please find the comments above

---

> ### Author Response · Authors · 2023-11-21
> **Response to Reviewer 2ic1**
>
> Thank you for providing such an encouraging and positive response. It is truly gratifying to know that you value the whole story of our paper, including our motivation, our proposed method, our theoretical analysis and our presentation. In light of your input, we are fully dedicated to addressing your concerns and offering comprehensive responses to the questions you have raised, as outlined in the forthcoming sections.
>
> > What is the intuitive reason to have double normalization in (12)?
>
> Thank you for your question.
> Previous work [1] have summarized the Temporal Covariate Shift (TCS), which means the distribution of the input current in each time-step are different.
> This difference will cause different mean and variance among time-steps.
>
> If we just take batch normalization on each time-step separately (refer to BN(Vanilla) in **Experimental settings** in **Response to All Reviewers** for details), it will cause the linear transformations in BN forward pass differ among time-steps. This difference in BN transformation will degrade the overall performance intuitively.
>
> As a result, we want a method that uses the same mean and variance to normalize layer input statistics each time-step. OSR accomplishes this task by applying the same BN forward transformation for each time-step.
> Additionally, the BN forward transformation is the same among the training and testing phase under OSR, so it helps improve the test accuracy.
>
>     [1] Duan, C., Ding, J., Chen, S., Yu, Z., & Huang, T. (2022). Temporal effective batch normalization in spiking neural networks. Advances in Neural Information Processing Systems, 35, 34377-34390.
>
>
> > I suggest to provide the experimental results to verify the Gaussion assumption.
>
> Thanks for your advice. In the main content, we make the assumption that the membrane potential have a Gaussian distribution. To verify the assumption, we collect the membrane potential of a VGG network trained with OSR and OTS on the CIFAR-10 dataset and visualize the distribution of membrane potentials for each layer and each time step. We display the results in Fig. 3 in the appendix of the revised paper. These distributions display the shape of bell curves, which similars to Gaussian distributions.
>
>
> > I did not find the calculation form of $\hat{\mu}$ and $\hat{\sigma}$. I suggest to provide a clear definition of them in the paper.
>
> Thank you for mentioning this. We have summarized the calculation of $\mu[t], \sigma^2[t], \hat{\mu}, \hat{\sigma^2}$ during training in **Response to All Reviewers**.
>
>
>
> > I did not find the meaning of m in (8).
>
> It means the number of elements in a group of normalization.
> For an image feature map at time $t$ of size $B \times C \times H \times W$ where $B$ is the batch size, $C$ is the number of channels, $H$ and $W$ are height and width, a group contains all elements in a channel, so $m = B \times H \times W$.

---

> > ### Comment · Reviewer_2ic1 · 2023-11-23
> > **Thank you for all clarifications**
> >
> > I have read the author's comment to all reviewers. I am satisfied with your answers and have no further questions. I remain with my recommendation.

---

### Official Review · Reviewer_9e6u · 2023-11-03

**Soundness:** 2 fair
**Presentation:** 2 fair
**Contribution:** 2 fair
**Rating:** 6
**Confidence:** 4

**Summary:**

The work focuses on online training techniques, especially addressing the limitation of not having access to future information in early time steps in online training. This paper tries to incorporate BN into online training, by two new modules, Online Spiking Renormalization (OSR) and Online Threshold Stabilizer (OTS).
This online training setting will benefit memory consumption. However, there are several problems:
the presentation of the new idea is not clear at all, it depends on the running mean and running variance, but how to calculate them is not mentioned clearly. we still don't know how to save memory by using the proposed modules.
The other issue is that in the theoretical parts, authors tend to make unrealistic assumptions. How can the spike train be iid. and expect the weights to be iid. Another thing is that the conclusions of the theorems, why do we need expectation of the average of u[t], and expectation of the average of sigma[t], how can these results help with the main new modules?
For the experiments, there is only one method using the same structure as the new method, shouldn't compared with different network architectures? Shouldn't compare with different BN methods? That is absent in the main paper.

**Strengths:**

This paper proposes a novel approach to training spiking neural networks that is memory-efficient and biologically plausible. The proposed Online Spiking Renormalization and Online Threshold Stabilizer techniques ensure consistent parameters and stable neuron firing rates across time steps.

**Weaknesses:**

The main idea should be better explained to make it easier to understand.
How to compute gradients in the backward stage if we use this forward transformation? The authors asked themselves the question, but did not give clear answers. Please make the core part of the paper clear.

**Questions:**

This online training setting will benefit memory consumption. However, there are several problems:
the presentation of the new idea is not clear at all, it depends on the running mean and running variance, but how to calculate them is not mentioned clearly. we still don't know how to save memory by using the proposed modules.
The other issue is that in the theoretical parts, authors tend to make unrealistic assumptions. How can the spike train be iid. and expect the weights to be iid. Another thing is that the conclusions of the theorems, why do we need expectation of the average of $u[t]$, and expectation of the average of $\sigma[t]$, how can these results help with the main proposed modules?
For the experiments, there is only one method using the same structure as the new method, shouldn't compared with different network architectures? Shouldn't compare with different BN methods? That is absent in the main paper.
How to calculate $\hat{u}, \hat{\sigma}$? please explain.
Is $\gamma$, $\beta$ learnable or fixed?
No point of Online Threshold Stabilizer (OTS) to stabilize the firing rate of each layer?
No experiment results on resnet on cifar datasets.

---

> ### Author Response · Authors · 2023-11-21
> **Response to Reviewer 9e6u (part 1)**
>
> Thank you for your constructive feedback. We are committed to clarifying addressing your concerns and make the presentation of our paper clearer, as outlined in the following sections.
>
> > How to compute gradients in the backward stage if we use this forward transformation? Please make the core part of the paper clear.
>
> Thank you for your question.
> In the following we provide a comprehensive description of the process of our algorithm:
>
> First of all, our algorithm works under the online learning framework, which means the network goes through forward and backward propagations step by step from time-step $1$ to $T$ (instead of first forward from time step $1$ to $T$ and then backward from time step $T$ to $1$).
> Since the network is processed step by step, it does not require saving intermediate state from time-step $1$ to $T$ as in regular BPTT.
> In each time-step, the information go from the input layer to the output layer of the network in forward pass, and then the gradients go from the output layer to the input layer in backward pass.
>
> **The process of the forward pass for each layer** are shown as follows (where the calculation of $\mu[t], \sigma^2[t], \hat{\mu}, \hat{\sigma^2}$ are shown separately in the following section):
> - Input: We need the output of last layer $s^{l-1}[t]$ (for input layer, this part is the input spike train / image at time $t$) and the weight between last layer and current layer $w^{l}$ ($s^{l-1}[t]$ and $w^{l}$ are both tensors instead of scalars).
> - Step 1: Calculate $I[t] = conv(s^{l-1}[t], w^{l})$ or $I[t] = linear(s^{l-1}[t], w^{l})$ or whatever else according to the layer type.
> - Step 2: Apply OSR on $I[t]$ to get the normalized $\tilde{I}[t]$:
>   During training, first we normalize $I[t]$ to $\hat{I}[t]$ (note $\hat{I}[t]$ passes gradients to $\mu[t]$ and $\sigma^2[t]$):
>   $$\hat{I}[t] = \frac{I[t] - \mu[t]}{\sqrt{\sigma^2[t] + \epsilon}}$$
> Then we apply linear transformation from $\hat{I}[t]$ to $\tilde{I}[t]$ (note $\tilde{I}[t]$ does not pass gradients to $\mu[t]$ and $\sigma^2[t]$ due to $no\\_grad()$, and $\gamma, \beta$ are learnable):
>   $$\tilde{I}[t] = \gamma \cdot \left( \hat{I}[t] \cdot no\\_grad\left( \frac{\sqrt{\sigma^2[t] + \epsilon}}{\sqrt{\hat{\sigma^2} + \epsilon}} \right) + no\\_grad\left(\frac{\mu[t] - \hat{\mu}}{\sqrt{\hat{\sigma^2} + \epsilon}} \right) \right) + \beta$$
>   As a result, $\tilde{I}[t] = \gamma \cdot \frac{ I[t] - \hat{\mu}}{\sqrt{\hat{\sigma^2} + \epsilon}} + \beta$ in the forward pass in training. Note that we apply conventional linear transformation $\tilde{I}[t] = \gamma \cdot \frac{ I[t] - \hat{\mu}}{\sqrt{\hat{\sigma^2} + \epsilon}} + \beta$ in the testing phase, so the input goes through the same linear transformations **among all time-steps in both training and inference**.
> - Step 3: Update membrane potential of neurons in layer $l$ according to the LIF neuron model and input $\tilde{I}[t]$: $$u^{l}[t] = (1 - \frac{1}{\tau^{l}}) u^{l}[t-0.5] + \tilde{I}[t]$$
> - Step 4: Apply OTS to update the threshold $\theta[t]$: $\theta[t] = \mu_\text{mem}[t] + \sigma_\text{mem}[t] \cdot \frac{\theta[1] - \mu_\text{mem}[1]}{\sigma_\text{mem}[1]}$
> - Step 5: Fire spikes $s^{l}[t]$: $$s^{l}[t] = \Theta(u^{l}[t] - \theta[t])$$ and then reset membrane potential: $$u^{l}[t+0.5] = u^{l}[t] (1 - s^{l}[t])$$
>
>
> The OSR forward transformation is shown in Step 2 in **The process of the forward pass for each layer**.
> The gradient for layer input, $\frac{\partial \mathcal{L}}{\partial I[t]}$ is thus
> $$
> \frac{\partial \mathcal{L}}{\partial I[t]} = \frac{\partial \mathcal{L}}{\partial \tilde{I}[t]} \cdot \frac{\partial \tilde{I}[t]}{\partial \hat{I}[t]} \cdot \frac{\partial \hat{I}[t]}{\partial I[t]}.
> $$
> The thing should be noticed is that $\tilde{I}[t]$ does not pass gradients to $\sqrt{\sigma^2[t] + \epsilon}$ while $\hat{I}[t]$ do.
> This is because $\sqrt{\sigma^2[t] + \epsilon}$ is in the function $no\\_grad$ when calculating $\tilde{I}[t]$, while it is not in $no\\_grad$ when calculating $\hat{I}[t]$.
>
> The remaining part is common: $\frac{\partial \tilde{I}[t]}{\partial \hat{I}[t]}$ is just $\gamma \cdot \frac{\sqrt{\sigma^2[t] + \epsilon}}{\sqrt{\hat{\sigma^2} + \epsilon}}$ and $\frac{\partial \hat{I}[t]}{\partial I[t]}$ can be calculated in the same way as batch normalization.
> Besides, the gradients to scaling factors $\beta, \gamma$ can be calculated in the common way:
> $$
> \frac{\partial \mathcal{L}}{\partial \gamma} = \sum_x \frac{\partial \mathcal{L}}{\partial \tilde{I}_x[t]} \left( \hat{I}_x[t] \cdot \frac{\sqrt{\sigma^2[t] + \epsilon}} {\sqrt{ \hat{\sigma^2} + \epsilon}} + \frac{\mu[t] - \hat{\mu}}{\sqrt{\hat{\sigma^2} + \epsilon}} \right),
> $$$$
> \frac{\partial \mathcal{L}}{\partial \beta} = \sum_x \frac{\partial \mathcal{L}}{\partial \tilde{I}_x[t]}.
> $$

---

> ### Author Response · Authors · 2023-11-21
> **Response to Reviewer 9e6u (part 2)**
>
> > The presentation of the new idea is not clear at all, it depends on the running mean and running variance, but how to calculate them is not mentioned clearly.
>
> Thank you for your constructive feedback. We have clarified the calculation of $\mu[t], \sigma^2[t], \hat{\mu}, \hat{\sigma^2}$ during training in **Response to All Reviewers**.
> If you have any further questions, please feel free to ask.
>
>
> > We still don't know how to save memory by using the proposed modules.
>
> Thank you for your question. Saving memory is a property of online training instead of our proposed modules.
> The reason that online training approaches can save memory compared with BPTT approaches is as follows:
> - BPTT methods require information of all time-steps in backpropagation since they need to backpropagate gradients from the last time-step to the first time-step. As a result, they need to save gradient information of all time-steps during training, which is $O(T)$ where $T$ denotes the total number of time-steps.
> - Online training algorithms backpropagates at each time-step, and they only require information at (or before) the current time-step. If information before the current time-step is required, they can be aggregated to some variables (called eligibility traces in prior works) costing constant memory. Overall, it only require $O(1)$ memory cost with respect to $T$.
>
>
> > Authors tend to make unrealistic assumptions. How can the spike train be iid. and expect the weights to be iid.
>
> It should be mentioned that BN itself relies on the fact that elements to be normalized are i.i.d samples from the training distribution [1][2].
> Moreover, some works analyzing BN-like mechanisms make assumptions of independence [3].
> As a result, our i.i.d assumptions are not that unrealistic.
>
>     [1] Ioffe, S. (2017). Batch renormalization: Towards reducing minibatch dependence in batch-normalized models. Advances in neural information processing systems, 30.
>     [2] Summers, C., & Dinneen, M. J. (2019). Four things everyone should know to improve batch normalization. International Conference on Learning Representations.
>     [3] Yan, J., Wan, R., Zhang, X., Zhang, W., Wei, Y., & Sun, J. (2020). Towards stabilizing batch statistics in backward propagation of batch normalization. International Conference on Learning Representations.
>
>
> > Another thing is that the conclusions of the theorems, why do we need expectation of the average of $\mu[t]$, and expectation of the average of $\sigma[t]$, how can these results help with the main proposed modules?
>
> Thank you for your question. The aim of our theoretical analysis is to show that our OTS helps stabilize our OSR mechanism. To achieve this goal, we show that OTS helps reduce the (expectation of) sample variance.
> This variance reduction will lead to more accurate estimation of running mean and running variance (for all time-steps) since it reduces the variance in estimation.
> In addition, this variance reduction come from the part of variance of mean among time-steps (lower 1st moment of distribution). This reduces the temporal covariate shift (which indicates the distribution difference) among time-steps.
>
>
> > For the experiments, there is only one method using the same structure as the new method, shouldn't compared with different network architectures? No experiment results on resnet on cifar datasets.
>
> We have conducted experiments of Resnet-19 on CIFAR10 and CIFAR100.
> The test accuracy for Resnet-19 is 95.20\% on CIFAR10 and 77.86\% on CIFAR100.
>
>
> > Shouldn't compare with different BN methods?
>
> We have listed the result of a vanilla BN which normalizes batch data at each time-step (you can go to **Experimental settings** in **Response to All Reviewers** for more details) testing on the Imagenet dataset in Table 1 (the line BN(Vanilla)).
> In addition, we have listed previous works of SNN BN methods (tdBN and TEBN) in Table 1. It should be noticed that these methods cannot be fitted into online training since they both require information of all time-steps.
>
>
> > How to calculate $\hat{\mu}, \hat{\sigma}$? please explain. Is $\gamma$, $\beta$ learnable or fixed?
>
> We have summarized the calculation of $\hat{\mu}$ and $\hat{\sigma}$ in the **The calculation of $\mu[t], \sigma^2[t], \hat{\mu}, \hat{\sigma^2}$ during training** part in **Response to All Reviewers**.
> In addition, $\gamma$ and $\beta$ are learnable.
>
>
> > No point of Online Threshold Stabilizer (OTS) to stabilize the firing rate of each layer?
>
> We have plotted the firing rate statistics of different configurations in Figure.2 (b). Results have shown that the firing rate across time-steps is stabilized when OTS is added.

---

> > ### Comment · Reviewer_9e6u · 2023-11-22
> >
> > After carefully reading the authors' responses, I have better understanding of the paper and my concerns have been addressed. I do not have new concerns, and I am more than willing to increase my score from 5 to 6.

---

### Author Response · Authors · 2023-11-21
**Response to All Reviewers**

We express our gratitude to the reviewers for their constructive feedback, which provides valuable insights for enhancing our paper. We will address any omissions in the original paper.
In this common response, we will clarify the calculation of $\mu[t], \sigma^2[t], \hat{\mu}, \hat{\sigma^2}$ during training and the BN(vanilla) model used as the baseline of our experiment.
<!-- In this section, we will provide a clarification of the entire process of our method and address important details that may have been omitted in the experimental setting in our original paper. -->





> **The calculation of $\mu[t], \sigma^2[t], \hat{\mu}, \hat{\sigma^2}$ during training** are shown as follows:

Firstly, they are all calculated in step 2 in the above process of forward pass.
- Additional input: Current time-step $t$. This is to determine whether we should initialize variables or calculate running mean/variance at the current time-step.
- Step 2.1: calculate the batch mean $\mu[t]$ and variance $\sigma^2[t]$ according to $I[t]$ (which is Eq.(8) in the original paper):
    $$ \mu[t] = \frac{1}{m} \sum_{x=1}^m I_x[t] $$ and
    $$ \sigma^2[t] = \frac{1}{m} \sum_{x=1}^m (I_x[t] - \mu[t])^2 $$     Here $m$ is the number of elements in a channel (which forms a group for normalization).
- Step 2.2: According to Eq.(9)(10), we need variables $\mu$ and $\sigma$ to accumulate total mean and variance. Before the first time-step, we initialize $\mu$ and $\sigma^2$ to $0$:
    $$\mu \leftarrow 0, \sigma^2 \leftarrow 0 \text{ if } t=1$$
Then we accumulate total mean and variance according to Eq.(9)(10):
    $$\mu \leftarrow \mu + \frac{1}{T} \mu[t], \sigma^2 \leftarrow \sigma^2 + \frac{1}{T} (\sigma^2[t] + \mu[t]^2)$$
- Step 2.3: Calculation of running mean $\hat{\mu}$ and running variance $\hat{\sigma^2}$ in time-step $T$ (the last time step).
    First adjust $\sigma^2$ according to Eq.(10):
    $$\sigma^2 \leftarrow \sigma^2 - \mu^2 \text{ if } t=T$$
    Then calculate running mean $\hat{\mu}$ and running variance $\hat{\sigma^2}$:
    $$ \hat{\mu} \leftarrow \hat{\mu} + (1-momentum) (\mu - \hat{\mu}), \hat{\sigma^2} \leftarrow \hat{\sigma^2} + (1-momentum) (\sigma^2 - \hat{\sigma^2}) \text{ if } t=T$$
    We take $momentum=0.9$ as in BN.


> Experimental settings:

We should clarify the baseline BN method (the line BN(Vanilla)) in Table.1 we select is almost the same with Reviewer W97A's Weaknesses 2 (ii):
The baseline vanilla BN integratings BN into the online training regime by calculating the statistics solely based on data from each time step.
In this approach, at every time step during training, the normalized input $\tilde{I[t]}$ is computed by
$$ \tilde{I[t]} = \gamma \cdot \frac{I[t] - \mu[t]}{\sqrt{\sigma^2[t] + \epsilon}} + \beta, $$
where the batch mean $\mu[t]$ and batch variance $\sigma[t]$ at time-step $t$ accords with **The calculation of $\mu[t], \sigma^2[t], \hat{\mu}, \hat{\sigma^2}$ during training** in the above (in Step 2.1).

The slight difference is the calculation of running mean and variance:
If we use the same momentum as in our OSR to update running mean and variance at each time step, they will be unstable since they are updated $T$ times more compared with our OSR.
A simple approach is to change the $momentum$ parameter to $1 - (1 - momentum) / T$, but we choose to implement it in a strict corresponding way:
We accumulate the mean and variance during training (for the variance, we accumulate by $\sigma^2 \leftarrow \sigma^2 + \frac{1}{T} \sigma^2[t]$ instead of the complex way shown above), and only update the running mean and variance at time step $T$.
In this way, there is no need to change the $momentum$ parameter.

---

### Meta-Review · Area_Chair_4PA5 · 2023-12-10

**Metareview:**

The reviewers were unanimous in their appraisal of this paper's contributions as above the bar for acceptance to ICLR.  I congratulate the authors on their detailed rebuttals, which were critical in leading some reviewers to raise their scores.  I'm pleased to report that this paper has been accepted to ICLR.  Congratulations!  Please revise the manuscript to address all reviewer comments and questions.

**Justification For Why Not Higher Score:**

The reviewers were unanimous in appraising this paper as above the bar for acceptance, with a final average score of 7.  They felt that it made novel theoretical as well as practical contributions, thus it seems like a solid case for a spotlight. (However, online stabilization of spiking networks may be a slightly niche topic, so I would be ok if the SACs / PCs decide it is better suited to be a poster).

**Justification For Why Not Lower Score:**

The reviewers were unanimous in appraising this paper as above the bar for acceptance, with a final average score of 7.  They felt that it made novel theoretical as well as practical contributions, thus it seems like a solid case for a spotlight.

---

### Decision · Program_Chairs · 2024-01-16

Accept (spotlight)